# AhR and CYP1B1 Control Oxygen Effects on Bone Marrow Progenitor Cells: The Enrichment of Multiple Olfactory Receptors as Potential Microbiome Sensors

**DOI:** 10.3390/ijms242316884

**Published:** 2023-11-28

**Authors:** Michele C. Larsen, Catherine M. Rondelli, Ahmed Almeldin, Yong-Seok Song, Alhaji N’Jai, David L. Alexander, E. Camilla Forsberg, Nader Sheibani, Colin R. Jefcoate

**Affiliations:** 1Department of Cell and Regenerative Biology, University of Wisconsin School of Medicine and Public Health, Madison, WI 53705, USA; mlarsen@medicine.wisc.edu (M.C.L.); almeldin@med.tanta.edu.eg (A.A.); 2Department of Biological Sciences, University of Delaware, Newark, DE 19716, USA; rondelli@udel.edu; 3Department of Ophthalmology and Visual Sciences, University of Wisconsin School of Medicine and Public Health, Madison, WI 53705, USA; song224@wisc.edu; 4Department of Pathobiological Sciences, University of Wisconsin, Madison, WI 53706, USA; njai@svm.vetmed.wisc.edu; 5Institute for the Biology of Stem Cells, University of California, Santa Cruz, CA 95064, USA; 2dwave@gmail.com (D.L.A.); cforsber@ucsc.edu (E.C.F.)

**Keywords:** CYP1B1, macrophages, bone marrow, olfactory receptors, oxidative stress

## Abstract

Polycyclic aromatic hydrocarbon (PAH) pollutants and microbiome products converge on the aryl hydrocarbon receptor (AhR) to redirect selective rapid adherence of isolated bone marrow (BM) cells. In young adult mice, Cyp1b1-deficiency and AhR activation by PAH, particularly when prolonged by Cyp1a1 deletion, produce matching gene stimulations in these BM cells. Vascular expression of Cyp1b1 lowers reactive oxygen species (ROS), suppressing NF-κB/RelA signaling. PAH and allelic selectivity support a non-canonical AhR participation, possibly through RelA. Genes stimulated by Cyp1b1 deficiency were further resolved according to the effects of Cyp1b1 and Cyp1a1 dual deletions (DKO). The adherent BM cells show a cluster of novel stimulations, including select developmental markers; multiple re-purposed olfactory receptors (OLFR); and α-Defensin, a microbial disruptor. Each one connects to an enhanced specific expression of the catalytic RNA Pol2 A subunit, among 12 different subunits. Mesenchymal progenitor BMS2 cells retain these features. Cyp1b1-deficiency removes lymphocytes from adherent assemblies as BM-derived mesenchymal stromal cells (BM-MSC) expand. Cyp1b1 effects were cell-type specific. In vivo, BM-MSC Cyp1b1 expression mediated PAH suppression of lymphocyte progenitors. In vitro, OP9-MSC sustained these progenitors, while Csf1 induced monocyte progenitor expansion to macrophages. Targeted Cyp1b1 deletion (*Cdh*5-Cre; *Cyp1b1^fl/fl^*) established endothelium control of ROS that directs AhR-mediated suppression of B cell progenitors. Monocyte Cyp1b1 deletion (Lyz2-Cre; *Cyp1b1^fl/fl^*) selectively attenuated M1 polarization of expanded macrophages, but did not enhance effects on basal M2 polarization. Thus, specific sources of Cyp1b1 link to AhR and to an OLFR network to provide BM inflammatory modulation via diverse microbiome products.

## 1. Introduction

The bone marrow (BM) is composed of a diverse population of cells that derive from hematopoietic and mesenchymal lineages. These cells interact with endothelial cells (EC) and pericytes at niche expansion sites in arterioles and sinusoids [1,2]. The expansion of B lymphocytes and the myeloid lineages occur clonally in such a niche [3]. Multipotential BM mesenchymal stem cells (BM-MSC) not only release support factors for each lineage, but also differentiate to adipocytes, osteoblasts, and chondrocytes [4]. Cyp1b1 expression in BM-MSC parallels the generation of hematopoietic regulatory factors [5]. Cyp1b1 also controls bone homeostasis [6]. Cyp1a1 and Cyp1b1 are each induced by the aryl hydrocarbon receptor (AhR), which is activated by both endogenous and exogenous substrates, thus engaging feedback control [7,8].

Benzo(a)pyrene (BP) metabolism by Cyp1b1 and Cyp1a1 generates different metabolites to suppress and then regenerate, respectively, lymphoid and myeloid progenitors within 6–24 h [9,10,11]. A subfraction of these diverse hematopoietic and mesenchymal cell types adhere to activated plastic, as described by many laboratories over past 20 years [12]. Importantly, we report here that this adherence is complete within 30 min, thus capturing much of the sinusoid expression. Over 24–72 h, these adherent cells are redirected by paracrine factors from non-adherent cells [13]. We searched for distinctive mechanistic features of this adhesion process by systematically changing the BM niche environment.

The Cyp1b1 gene exhibits many distinctive features compatible with developmental and endocrine regulation [14,15]. For example, there is novel control through elements in the conserved, long 3′-UTR [16,17]. Cyp1a1 lacks such features, and typically has minimal expression without external AhR activation. This stimulation is delivered through canonical signaling, which involves ligand activation of a cytoplasmic complex of AhR with heat shock protein 90 (HSP90), the small modulator, p23, and AIP/Ara9 (AhR-interacting protein/AhR-associated protein 9) [18]. This activation delivers AhR to the nuclear partner, ARNT (AhR nuclear translocator), at specific promoter elements (drug response element: DRE). This AhR complex delivers environmental modulation of hematopoietic and inflammation networks [19,20,21].

This AhR/CYP1 partnership functions within a system that extends from BM vascular arterioles and sinusoids to the gut epithelium via arterial blood flow [18]. The access of microbiome metabolites to BM is determined by their transfer across the intestinal epithelial barrier [19]. AhR induction of Cyp1a1 in gut epithelia is extensive but absent in germ-free mice. However, Cyp1b1 expression remains substantial, consistent with constitutive AhR control [20]. Specific microbiome metabolism delivers the AhR inducers required for this Cyp1a1 expression. Thus, many of the polycyclic products from tryptophan metabolism or dietary indoles are Cyp1 substrates [19,22,23].

BM activation is also delivered by small microbiome fat metabolites, including short-chain fatty acids. BM cells appear to adapt to such metabolites through a large set of G-protein-coupled receptors (GPCR) that have been repurposed from the olfactory bulb receptors (OLFR) [23,24]. Previous work has identified OLFR participation as single entities. For example, Olfr821 responds to octanoic acid to function in pancreatic beta cells [25]. In addition, complementary activity is provided by Cyps, which metabolizes fatty acids such as the Cyp4f and 2j forms [20]. Cyp1b1 also metabolizes polyunsaturated fatty acids to epoxides and HETE, which modulate blood flow [26]. We show here that the largest cluster of Cyp1/AhR changes in BM substantially affect these OLFR. Such microbiome generation of fatty acids has been linked to energy homeostasis [27].

The BM vasculature typically has low oxygen concentration in the sinusoids that increases with proximity to the arterioles or blood flow [18]. Cyp1b1 in vascular EC and pericytes promotes cell adhesion by suppressing reactive oxygen species (ROS) affecting their proangiogenic activity when oxygen levels rise above two percent [28]. Induction of Cyp1a1 generates ROS through an uncoupling of the NADPH reduction cycle [29]. The lung represents a tissue in which coordination of Cyp1b1 with the induction of Cyp1a1—by AhR—are central features of ROS regulation [30,31,32]. In addition, the kynurenine pathway—initiated by AhR-inducible indole di-oxygenase [33] —produces not only AhR ligands, but also quinolinic acid, an iron chelator that enhances ROS production [34,35]. Global deletion of Cyp1b1 suppresses the hepatocyte release of hepcidin, resulting in increased circulating iron availability through increased levels of ferroportin (Slc40A1) in macrophages and enterocytes. Vascular cells that express ferroportin, such as retinal EC, demonstrate elevated intracellular iron and ROS [36,37].

The adhesion of such BM cells is altered by both in vivo Cyp1b1 deletion and 2,3,7,8 -tetrachlorodibenzo-p-dioxin (TCDD) activation of AhR. BM cells exhibit distinctive responses under the control of growth hormones in young mice. The mice used for these studies are C57BL/6J at an age range of 5–7 weeks. Gene expression analyses were limited to female mice. Here, we demonstrate parallel overlapping stimulations of gene expression. Cyp1a1 deletion has much less of an effect. These Cyp1b1-AhR changes show non-canonical features that could arise from ROS participation, mediated by NF-κB [38,39,40,41] or Nrf2 partnership with AhR [42,43,44,45]. The relationship between Cyp1b1 and AhR in these adherent BM assemblies has been systematically addressed by examining the gene expression effects of in vivo Cyp1 deletions (1a1^-/-^, 1b1^-/-^, and dual deletions 1a1^-/-^;1b1^-/-^ (DKO)). Acute AhR activation is compared for metabolism resistant, TCDD, and different PAH exposures, including with the deletion of Cyp1 forms [41,42,46]. A set of twelve conditions, applied to individual mice, effectively resolved gene clusters that function in adherent BM cells with coordinated control by Cyp1b1, Cyp1a1, and AhR.

Extensive overlap of *Cyp1b1* deletion and AhR regulatory functions in these adherent assemblies reveals cells that exhibit novel signaling through the POL2-A subunit of RNA polymerase 2 (Polr2a gene). We show that this catalytic subunit changes expression, independent of its remaining 11 subunits [41,42]. Close correlations between expression levels of Polr2a and 20 BM OLFR, across multiple treatments, were identified. In parallel with the resolution of these novel gene expression pathways in adherent BM assemblies, we examined the functional impacts of Cyp1b1 on the lymphocytes and monocyte progenitors that equilibrate on and off the assemblies. Cyp1b1 has multiple functions that depend on the site of expression [43,47]. To explore such functions in freshly isolated BM cells, we have developed two in vitro models that ultimately utilize *Cyp1b1^fl/fl^* and targeted Cre deletions to compare general and lineage-selective contributions [48].

The strong coupling of Cyp1b1 to ROS suggests important roles in inflammation, including DNA damage control and macrophage polarization. We also explored Cyp1b1 regulation of ROS-induced DNA double strand breaks in BM lymphopoiesis originating from common lymphoid progenitors (CLP). Comparisons were made to the deletion of Xeroderma Pigmentosum Group C (XPC), a protein that initiates DNA repair [45]. Lineage effects of Cyp1b1 deletion characterizes a mechanism for selective endothelial cells (EC) support for lymphoid progenitor proliferation [49]. In BM, in vivo monocyte expansion to macrophages is driven by local EC release of colony stimulating factor 1 (Csf1), which activates the Csf1 receptor (Csf1R) on the progenitors [50]. In our second model, isolated BM cells were expanded to macrophages, in vitro, by Csf1. The appreciable Cyp1b1 expression in CD14^+^ monocytes [47,51,52] is mediated by interferon regulatory factor 4 (Irf4), a macrophage M2-polarizing factor [48]. Macrophage polarization was used to quantify this intervention. These in vitro models emphasize how Cyp1b1 produces diverse effects according to BM expression in vascular cells, monocytes, or mesenchymal cells. A partnership of Cyp1b1 with AhR in the control of local ROS and microbiome metabolites will depend on these in vivo origins.

## 2. Results

### 2.1. Rapid Cell Assembly of Isolated BM Cells, In Vitro, Is Similarly Affected by AhR Activation and Cyp1b1 Deletion

#### 2.1.1. Adherence of a Small Sub-Fraction of BM Cells

The adherence of mixed cell populations eluted from BM is characterized in many papers [1,2,4,5,12], including in our previous work [9,10,52]. These eluted cells represent populations in the BM sinusoids rather than bone matrices [6]. For WT C57BL/6J mice, only about five percent of the eluted cells adhere. Typical studies [1,12] focus on the cultures beyond 24 h when the adherent mesenchymal progenitors expand. After two weeks, these MSC dominate the culture. The BMS2 cell line, which is used as a reference cell in the studies presented here, emerges from these primary cultures [9]. Here, we focused on cells that adhere within 30–60 min in order to optimally capture the in vivo mRNA expression. Their non-adherent counterparts were also examined. In gene expression array analyses described here, isolated mRNA from adherent cells from diverse treatments (Cy3 labeled) is compared to a constant reference comprising equal amounts of the same mix of cells from non-adherent WT cells (Cy5 labeled).

For WT mice, a reproducible adherent population was captured in this early window that did not increase as a proportion over the next 24 h. Based on the expression of cell-type gene markers, the dominant cell types in adherent WT cells were B lymphocytes, erythroblasts, and macrophages. The B cells and erythroblasts were elevated relative to non-adherent cells. Adherent MSC comprised less than one percent of the total. The much larger size of the fibroblastic cells enhances their role. A model for larger adherent assemblages anchored by fibroblastic MSC or osteoblasts is shown in Figure 1A. The activation of AhR with TCDD and the deletion of Cyp1b1 each change cell adhesion. They also reveal major increases in gene expression in the adherent MSC with novel characteristics linked to adhesion. Changes in gene expression for these rapidly adhering cells can scarcely be produced in 30 min, and therefore correspond to prior changes in the BM sinusoids.

Systematic analysis of the effects of Cyp1 deletions (*Cyp1b1*^-/-^, *Cyp1a1*^-/-^, and both; *DKO*) and diverse AhR activations (TCDD vs. PAH, 6–24 h) established the overlapping effects of Cyp1b1 deficiency and AhR activation on genes in these assemblies (Figure 1B and Figure 2A). To resolve the effects of *Cyp1b1* deficiency, which occur exclusively with BM cells in vitro, we developed models that depended on changes in lymphocyte and monocyte progenitors (Figure 1B). The lymphocyte model depends on DNA damage incurred from ROS and PAH chemical stress over 24 h. Cyp1b1 effects are indirectly produced from cells surrounding the progenitors. The monocyte model targets the expansion of progenitors that express Cyp1b1. Changes in the macrophages can derive from both direct and indirect Cyp1b1 effects. We used lineage-selective deletions of Cyp1b1 to resolve the source-selective effects of Cyp1b1.

#### 2.1.2. Characterization by Flow Cytometry and Adhesion

Flow cytometry analysis of the freshly isolated BM cells showed selective effects of Cyp1b1 deficiency on the expansion of hematopoietic lineages (Figure 1C). BM cells from *Cyp1b1*^-/-^ mice produced direct changes in the steady state levels of progenitor cells [53] (Figure 1C). Lymphoid and myeloid progenitor colony forming unit (CFU) assays showed an equal distribution between adherent and non-adherent cells (Appendix A). The total number of BM cells was not significantly different between WT control and *Cyp1b1*^-/-^ mice or in WT animals after 24 h of TCDD treatment. However, in each case, adherence decreased two-fold (*p* ≤ 0.02) (Figure 1D). This adhesion process is highly sensitive to mouse strain, sex, age, and genetic background (6J vs. 6N) [54] (Appendix A). Because of this complexity, this report used precise breeding with defined diets, examining BM cells of young C57BL/6J females (5–7 weeks). We systematically addressed adherent and non-adherent WT cells from control mice, and then adherent cells from 10 additional in vivo treatments (three individual mice examined separately in most groups). The adherent cells included about 30 percent macrophages, equally distributed between adherent and non-adherent populations. Histology markers (F4/80, CD80 and M2-selective CD206) each showed an enhanced presence after TCDD treatment. There was no difference in macrophage proportions between WT and *Cyp1b1*^-/-^ mice (Figure 1E).

**Figure 1 ijms-24-16884-f001:**
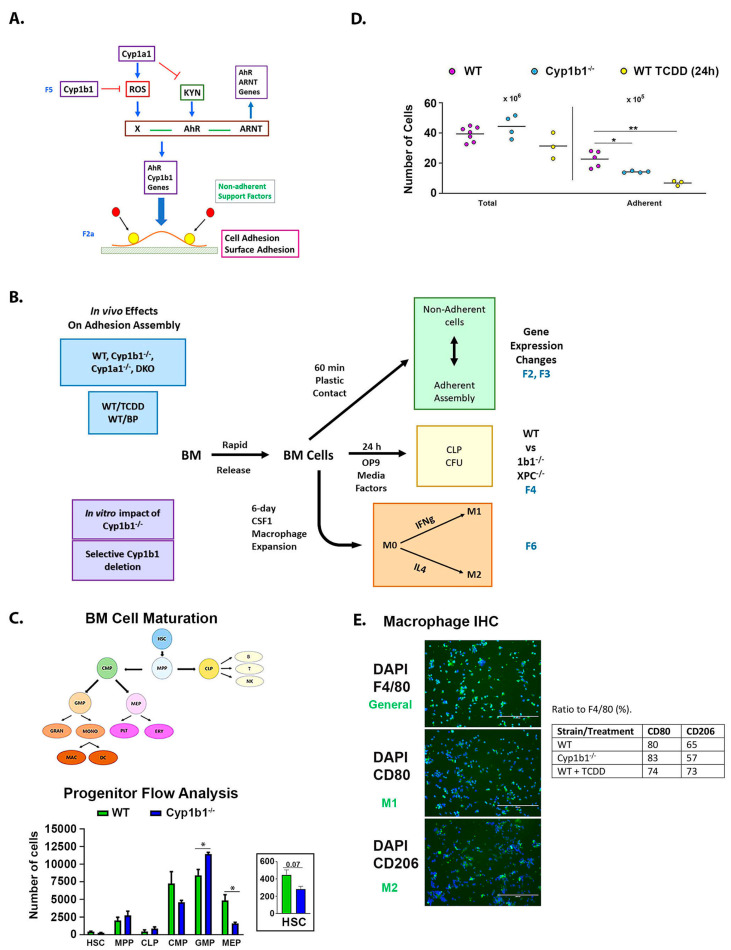
Effects of *Cyp1b1*^-/-^ and AhR activation on adherent BMC. (**A**) Model for adherence assembly of isolated BMCs. Overlapping in vivo effects of AhR and Cyp1b1 control content and adhesion signaling. Key features: Primary surface adhesion recruits additional cell types to multicell adherence assembly (Adh) that forms within 30 min. Factors from non-adherent (NAdh) cells impact the assembly. AhR is activated both by ROS-mediated processes and canonically by constitutive kynurenine products (KYN) arising from gut tryptophan metabolism. (**B**) Experimental strategy to probe *Cyp1b1*^-/-^ effects on adherent BMCs involves four approaches: Microarray analyses of gene expression changes caused by Cyp1 deletion and various AhR activation processes (Figure 2 and Figure 3). In vitro proliferation of lymphocyte progenitors (Figure 4 and Figure 5). In vitro expansion of myeloid progenitors to macrophages (Figure 6 and Figure 7). In vitro analyses allow resolution of Cyp1b1 effects from distinctive cell sources (selective Cre deletions) (Figure 4 and Figure 7). (**C**) Flow Cytometry resolves *Cyp1b1*^-/-^ effects on lymphoid and myeloid progenitors in isolated BMC (CLP, MMP, GMP, and CMP). *Cyp1b1*^-/-^ decreases HSC, CMP and MEP, while increasing GMP. (**D**) Adh cell numbers are suppressed in *Cyp1b1*^-/-^ mice and in WT mice after 24 h of TCDD treatment. Female mice (aged 5–7 weeks) are used here and for expression studies in Figure 2 and Figure 3. Results for male mice are shown in Appendix A. (**E**) Probe of macrophages in Adh assembly. DAPI stains all nuclei; specific antibodies resolve functional macrophage markers; F4/80: pan macrophage; CD80: M1 macrophage; CD206: M2 macrophage. DAPI staining of all cells shows both clusters (>10 cells) and individual cells. CD206 fluorescence (and F4/80) show that macrophages are heterogeneously distributed among total cells (DAPI). Integrated fluorescence intensities showed significant TCDD stimulation of the M2/CD206 macrophage, which is the prime contributor to F4/80 general increase. *Cyp1b1*^-/-^ has no effect. CLP: Common lymphoid progenitor; MMP: Monocytic myeloid progenitor; GMP: Granulocyte monocyte progenitor; CMP: Common myeloid progenitor; HSC: Hematopoietic stem cells; MEP: Megakaryocyte erythrocyte progenitor. * *p* < 0.05; ** *p* < 0.01.

### 2.2. Functionally Distinct AhR/Cyp1 Combinations Are Resolved by LIMMA Expression Analyses, Using a Multi-Treatment Matrix

#### 2.2.1. Design of a 12-Treatment AhR-Cyp1 Response Matrix for Adherent BM Cells

Adhesion, Cyp1 deletion and AhR activation gene responses were compared for 30 individual mice subjected to a set of different treatments (12-treatment matrix; Figure 2A). The time dependence of PAHs (6–24 h) extends across the hepatic clearance (75 percent within 6 h) [52]. Cyp1a1 deletion slows clearance, thus enhancing direct AhR activation while decreasing reactive metabolites. Response patterns were highly conserved within the 12 treatment groups, according to the mechanisms of interaction among AhR and the two forms of Cyp1. Group data for the adherent cells from mice in each treatment group (N = 3) were compared using Limma statistical analyses to the equivalent WT cells [54,55] (Appendix A). The response characteristics that distinguish each mechanism are shown within the Treatment Matrix of Figure 2A.

**Figure 2 ijms-24-16884-f002:**
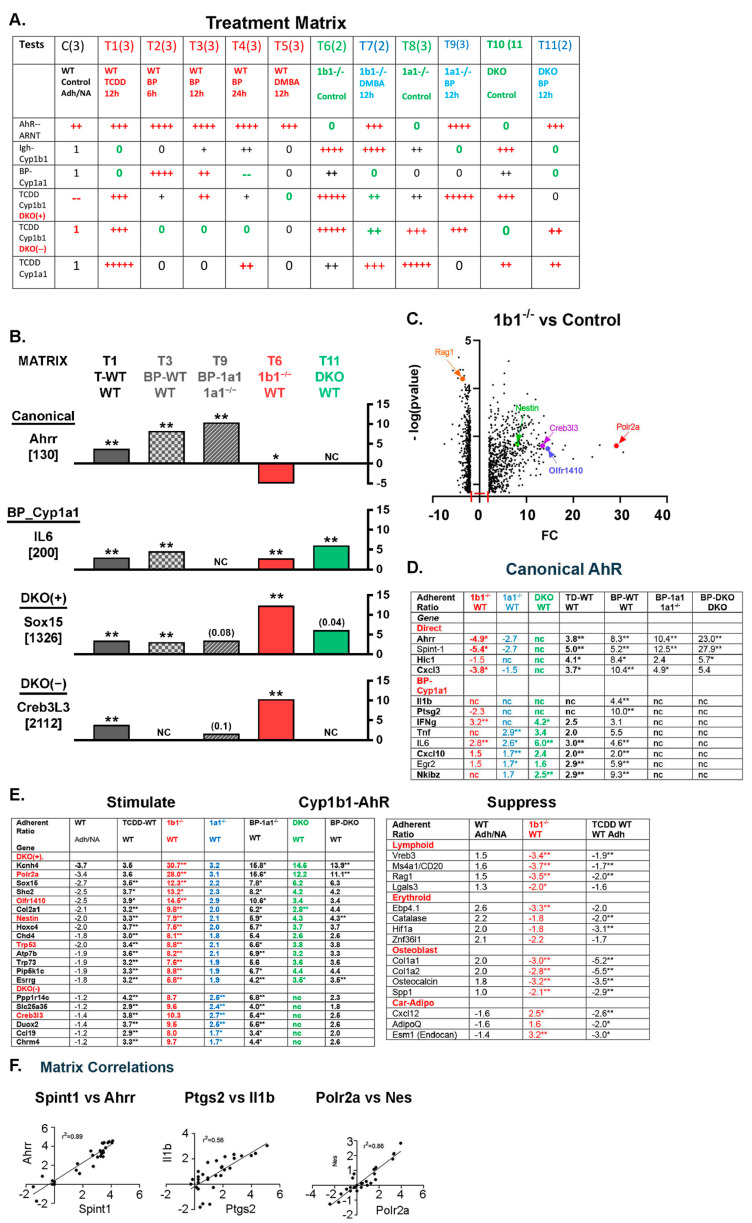
Resolution of multiple mechanisms of overlap between AhR treatments and Cyp1b1 deletion. Identification of a dominant Cyp1b1-AhR dual response process. (**A**) Treatment matrix for micro-array analyses. The resolution of WT BM cells into adherent and non-adherent together with analyses for adherent cells from eleven different treatments applied to female 5–7-week-old mice. The data represent an accumulation of experiments with individual mice over 12 months, thus introducing the variance of litter diversity. Each analysis is limited to 2–3 mice with different comparisons. Repeats provided three mice in each treatment (T) group, except for DKO mice (n = 2; T 10 and T 11), all mice were included. The gene expression array data derives from individual mice (see Appendix A). The output of each determination is a Cy3/Cy5 ratio corresponding to the treated mouse RNA (Cy3) and an equal addition of RNA from the same non-adherent cell mix. The treatment groups each analyzed adherent assemblages. They comprised a 12 h TCDD treatment (AhR activation only, T1), a time course for BP treatment (6 h, 12 h, 24 h; T2-T4), and one time point 12 h DMBA treatment (T5). BP and DMBA are PAH that both activate AhR and form reactive metabolites. Deletions of Cyp1b1 (T6), deletions of Cyp1a1 (T8), and the dual deletion (DKO) (T10) each matched to WT mice. The Cyp deletions were also compared to 12 h PAH treatments equivalent to those used for WT mice. *Cyp1b1*^-/-^ DMBA (T7), *Cyp1a1*^-/-^ BP (T9), and DKO BP (T11). The matrix set of treatments distinguishes different gene response mechanisms (**B**). The data were processed by a LIMMA statistics algorithm applied to a triplicate set to provide ratios from a two-way comparison of treatment groups. Mostly, the typical comparison refers to WT adherent. Cy3 levels provide a guide to expression levels. False positives were minimized by using a Cy3 cut-off of 200 for the higher of WT or by treating. Positive ratios apply to fold stimulations relative to WT (or designated reference). Negative ratios apply to fold suppressions. LIMMA ** *p* < 0.01 and * *p* < 0.05. (**B**) Application of treatment matrix. Five matrix treatments compared to appropriate reference treatments showed distinctive patterns corresponding to different gene activation mechanisms. Each mechanism class is represented by one representative gene. The relative basal expression is shown (Cy3). Canonical AhR activation (Ahrr) derives directly from the AhR-ARNT heterodimer, enhanced by direct TCDD or BP binding. Deletions of Cyp forms can point to endogenous ligands, but minimal responses were shown here. The BP-Cyp1a1 response depends on a BP metabolite. In (**D**), IL1β and Ptgs2 respond only to BP in WT via Cyp1a1, but not in Cyp1a1^-/-^ and not to TCDD (metabolism resistant AhR ligand). IL6 utilizes the BP metabolite (WT-BP but not *Cyp1a1*^-/-^ BP). A novel dominant mechanism for these assemblages (designated Cyp1b1-AhR) is indicated by parallel stimulations from *Cyp1b1*^-/-^ and direct AhR activation by TCDD (*Cyp1b1*^-/-^ >> TCDD). This dual selectivity is shown by mesenchymal development mediators, Sox15 and Creb3L3. A major subset (80 genes) represented by Sox15 also showed stimulation in the DKO. A smaller subset represented by Creb3L3 had no stimulation in DKO assemblages. We distinguished these subsets in (**E**) as DKO(+) and DKO(-) genes, respectively. The additional TCDD, *Cyp1b1*^-/-^, and DKO changes seen for IL6 revealed a DKO(+) contribution not seen for IL1β and Ptgs2. (**C**) Volcano plot of LIMMA expression ratios (fold change; FC) plotted against -log *p*-values for Adh *Cyp1b1*^-/-^ vs. Adh WT. Plot limits: expression WT > 300; FC > 2.0; ** *p* < 0.01. (**D**,**E**) Cyp1b1-AhR genes; DKO(+), DKO(-) clusters; Stimulations/Suppressions shared by *Cyp1b1*^-/-^ and AhR activation (Complete listing, Appendix A). (**F**) Genes responding to a dominant Cyp1/AhR process were highly correlated for treatment matrix responses (r^2^ = 0.5–0.9: Ahrr/Spint1; Ptgs2/IL-1β and Nestin/Polr2a). * *p* < 0.05; ** *p* < 0.01.

The response patterns link to different lineages identified in the adherent cluster. Figure 2B shows a set of distinct gene response patterns derived from the matrix treatments. (see legend). Ahrr provides a canonical response pattern typical of direct activation of the AhR-ARNT complex. IL6 shows BP stimulation in WT mice, but no response to BP in *Cyp1a1*^-/-^ mice, a combination delivers maximum BP canonical activation of AhR. This combination typifies the BP-Cyp1a1 mechanism, where the peak BP stimulation occurs within 6h (matrix T2), and unmetabolized TCDD only contributes through other mechanisms. Such mixed responses are common. Positive responses to *Cyp1b1*^-/-^, DKO, and TCDD are not seen for either canonical stimulation or BP-*Cyp1a1*^-/-^. However, the mesenchymal development mediators Sox15 and Creb3l3 exhibit the Cyp1b1-AhR dual stimulation pattern (Figure 2B). Sox15 but not Creb3l3 was also increased after the dual deletions of *Cyp1b1* and *Cyp1a1* (DKO). Among the Cyp1b1-AhR genes, this stimulation in DKO BM resolves about 80 DKO(+) genes from 21 DKO(-) genes (Examples: Figure 2E/left). Suppressions shown as a narrow band in the Cyp1b1^-/-^ volcano plot (Figure 2C) mostly arise from net cell losses from the adherent assemblages. (Examples; Figure 2E/Right).

Erythroblasts, monocytes, lymphocytes, and BM-MSC in various states of differentiation have been identified, and changed their proportions accordingly with in vivo treatments. The AhR and Cyp1 participation in each cluster is explained in the Appendix A. Commonly, genes show mixed responses, indicative of expression in multiple cell types. The largest participation of AhR and Cyp1 involves a novel, dual, non-canonical activation by Cyp1b1 deletion and an optimal AhR activation by BP in *Cyp1a1*^-/-^ mice. The low levels of AhR in BM cells suggests proximal participation from the gut epithelium, where expression is the highest and the arterial connection is rapid [56].

#### 2.2.2. Control of Adhesion Assemblages by Cyp1b1

Comparison of genes differentially expressed in adherent cells from *Cyp1b1*^-/-^ mice with those in WT cells showed about 900 increases and decreases total. All expression determinations were normalized to the same Cy5 reference, comprised of mRNA from non-adherent WT cells. A plot of fold-change (FC; *Cyp1b1*^-/-^/WT) vs. -log P is shown in Figure 2C. Each expression ratio was determined as the mean expression level for each treatment group, derived from the LIMMA statistics. To simplify the display, we applied limitations on FC and expression level (FC > 2-fold, ** *p* < 0.01, up or down). The display showed fundamental differences between stimulation (broad scatter, FC 2- to 30-fold) and suppression (compressed to 2- to 4-fold). An examination of TCDD-stimulated WT cells vs. the same control WT cells reproduced a similar suppression pattern, which was compressed to a narrow range (2.5- to 4-fold).

Remarkably, a detailed examination of the differentially expressed genes showed very extensive overlap. Cyp1b1 deficiency and TCDD each selectively decreased the total adherence of cell types in the assemblage (2- to 3-fold; Figure 1D). This narrow range of expression decreases is compatible with decreases in the proportions of specific cell types, notably the substantial contribution of B cells. A complementary increase occurred from enrichment of the retained cell types (MSC). The broader range of stimulations from Cyp1b1 deficiency is indicative of a superposition of large stimulations on cell enrichment.

#### 2.2.3. Canonical AhR Activation vs. Cyp1a1-BP Response

The direct canonical AhR participation with ARNT is characterized here by BP and DMBA in C57BL/6J mice, through the AhRb allele. The AhRd allele, which prevails in many other strains [55], forms the heterodimer with ARNT, but with lower affinity for TCDD and complete resistance to PAH [57,58,59]. Here, we tested for a canonical partnership by additionally using a congenic C57BL/6J strain with the AhRd allele (Appendix A). Many other partners for AhR have been identified, notably from ROS signaling processes [60,61]. The AhRd mice can function with BP if a ligand is either not required or depends on a different active conformation [62].

In the adherent assemblies, only 11 genes exhibited canonical AhR/ARNT stimulations, each at very low expression levels. The second indirect cluster, Cyp1a1-BP, requires canonical induction of Cyp1a1 and specific BP metabolites [63,64,65]. These responses (20 genes) were characterized by a BP stimulation that peaks in 6 h and is lost after 24 h. *Cyp1a1*^-/-^ mice do not respond, indicative of mediation by metabolites. Neither TCDD nor DMBA (7,12-dimethylbenz[a]anthracene) are activators. This combination suggests mediation by a particular type of BP metabolite (primary quinone).

TCDD, BP, and DMBA generated similar BM AhR canonical activations in WT mice and after Cyp1 deletion (Figure 2B; upper, black). Constitutive canonical responses after Cyp1 deletions would mark an accumulation of AhR ligands, which are absent here (Figure 2B; red, blue, green; Appendix A). In the WT adherent cells, 8/11 canonical responders showed substantially elevated expressions relative to the non-adherent cells. Late erythroblast markers, which show similar selectivity, are also AhR-sensitive [49].

Twenty-seven cytokines include many like IL6 that show the BP-Cyp1a1 mechanism but with contributions from alternative mechanisms. The mechanism perhaps diversely reflects different inflammatory sources. The BP-Cyp1a1 is most specifically shown by IL1β and Ptgs2. It is a principal mechanism for Tnf and five other cytokines. Cxcl2 and Cxcl3 show canonical and BP-Cyp1a1 contributions. Ccl19 and less expressed IL2, IL15, and IL27 predominantly show optimum stimulations from Cyp1b1^-/-^ and BP-Cyp1a1^-/-^, respectively. These stimulations match the Cyp1b1-AhR patterning of Sox15. IL10 repeats the IL6 dual stimulation (Figure 1B).

Two more selective mechanisms have not been characterized for AhR participation. Mechanism-based expression changes can discriminate between Cyp1b1 or Cyp1a1 (Appendix A). Heavy immunoglobulins (IgH and IgJ) show DNA double strand break (DDSB) rearrangements in lymphocytes [66]. The large, selective basal stimulation by Cyp1b1 deficiency is consistent with an enhanced susceptibility to ROS, but not the effect of TCDD. The heat shock protein a1 (Hspa1; HSP70) generates a strong selective dependence on Cyp1a1 deficiency that is matched by high TCDD stimulation. The BP stimulation is less and only significant after 24 h. Tnf additionally shows a contribution that matches that of Hspa1. A matching selectivity for the loss of a complete set of highly expressed ribonuclease (eosinophil associated ribonuclease; EAR) forms indicates a loss of granules from eosinophils.

#### 2.2.4. Cyp1b1-TCDD Non-Canonical Stimulations Predominate as DKO(+) and DKO(-) Clusters

Among *Cyp1b1*^-/-^ AhR stimulations, the 80 genes of the dominant DKO(+) cluster showed stimulations relative to WT ranging from 5- to 30-fold. Each is linked to a negative adhesion factor (Figure 2E; left upper). The DKO(-) cluster showed relative constant 10-fold stimulations by *Cyp1b1*^-/-^ with no adhesion bias. The BP-*Cyp1a1*^-/-^ stimulations were constant, thus paralleling the distinctive constancy of the *Cyp1b1*^-/-^ stimulations and adhesion.

Olfr1410 appears among the most highly stimulated DKO(+) genes (Figure 2E, left). Many more OLFR were found at lower stimulations. The mRNA encoded olfactory neuron receptors (Appendix A), representatives of this largest family of mouse/human genes. They are often seen in small numbers in nonneuronal cells outside the olfactory bulb [23]. However, the almost 40 OLFR collected in these adherent BM cells represents a novel abundance that may reflect a functional re-purposing of these receptors for BM regulation.

#### 2.2.5. Cyp1b1 Deficiency-TCDD Suppressions Match Cell-Selective Adherent Losses

Cyp1b1 deficiency and TCDD treatment cause suppressions in equal number to the stimulations, although the volcano plot showed a narrow range of suppressions in the 2- to 4-fold range (Figure 2E, right panel). These suppressions include markers of several cell types that are both abundant and enriched in the WT-adherent assemblages: B-lymphocytes [50], late erythroblasts [67], and osteoblasts [68] (Figure 2E, left panel; Appendix A). Notably, among these suppressed genes are similarly expressed functional lymphocyte genes (Vpreb3, Ms4a1, Rag1), late erythroblast markers (Epb4.1, Catalase, Hif1a, Znf36l1), and the secreted bone and immune regulator (Spp1/osteopontin), which is highly expressed in osteoblasts [61]. Markers for Cxcl12-abundant reticular (CAR) cells also share selective suppression by TCDD.

#### 2.2.6. Matrix Correlation Plots Define Diverse AhR and Cyp1 Participations

Coherent gene clusters like canonical AhR or DKO(+) can be defined by expression correlations across the full matrix for 30 mice and 12 treatments. We see r^2^ values in the range of 0.7–0.9 (Figure 2F; Appendix A). Deviations capture participation of additional mechanisms. The most substantial direct canonical responses (Ahrr and Spint/Hai) are more closely matched than the Cyp1a1-BP processes for Ptgs2 and IL-1β (Figure 2B). Nestin and Polr2a, from the DKO(+) cluster, are highly correlated (Figure 2E). OLFR genes mostly contribute to DKO(+) and exhibit correlation between *Cyp1b1*^-/-^ and Cyp1a1-BP expression (Figure 3E).

**Figure 3 ijms-24-16884-f003:**
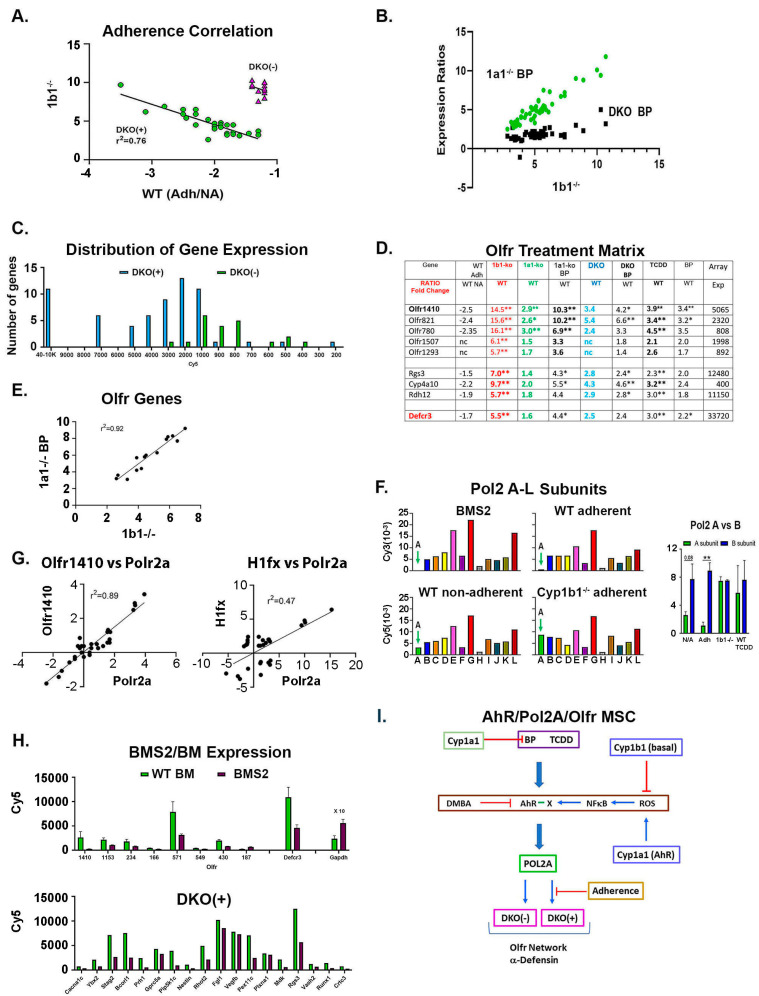
Novel features of Cyp1b1-AhR gene clusters. Four features were examined: the characteristics of DKO(+) and DKO(-) sub-clusters (**A**–**C**), the expression of Olfr genes (**D**,**E**), regulation through Pol2 subunit A (**F**,**G**), and the overlap with expression in BMS2 cells (**H**). (**A**) Cyp1b1-AhR genes in the DKO(+) cluster show a negative correlation of *Cyp1b1*^-/-^ stimulations with adherence, measured as WT (Adh/NA) (Figure 2D, Appendix A). (**B**) For Cyp1b1-AhR genes in the DKO(+) cluster, *Cyp1a1^-/-^* BP stimulations correlate with *Cyp1b1*^-/-^ stimulations and with lower DKO BP stimulations. (**C**) DKO(+) cluster includes highly expressed genes, indicative of expression in high abundance cells. DKO(-) genes show no equivalent high expression (Appendix A). (**D**) Olfactory receptors (OLFRs) link to α-Defensin 3. A total of 44 OLFR genes are distributed across DKO(+) and DKO(-) clusters (Appendix A). * *p* < 0.05; ** *p* < 0.01. (**E**) Highly expressed OLFR genes also show correlation of *Cyp1b1*^-/-^ with *Cyp1a1^-/-^* BP treatments (compare 3B). (**F**) Variability of RNA Polymerase 2 A subunit (Polr2a) (A- arrow) relative to other 11 subunits in BMS2 cells and adherent BM cells. Selective stimulation by *Cyp1b1*^-/-^. (Right) Variable Polr2a compared to constant Polr2b for activations involving *Cyp1b1*^-/-^. (**G**) Treatment matrix correlations for Polr2a with Olfr1410 and H1Fx, respectively. Matched *Cyp1b1*^-/-^ increases (red). *Cyp1a1*^-/-^ responses were scarcely significant (blue). Figure 2D also resolves these genes into two distinct clusters based on DKO stimulations (green). DKO(+) (upper) and DKO(-) (lower) were separated based on expression levels after the further deletion of Cyp1a1 (*Cyp1b1*^-/-^ to *DKO*). Twenty-one genes retained smaller but significant stimulations in cells from DKO(-) mice that paralleled those in *Cyp1b1*^-/-^ mice (DKO(+) genes). The appreciable loss of stimulation compared to the *Cyp1b1*^-/-^ level suggests a positive contribution by Cyp1a1 to *Cyp1b1*^-/-^ stimulation. (**H**) Compare Adh BM cells with BMS2 cells for the expression of multiple OLFRs, α-defensin (Upper), and other select DKO(+) genes (lower). (**I**) AhR model for Cyp1b1-AhR clusters within adherent MSC in Adh cell clusters (Figure 1A). A non-canonical AhR complex (AhR-X) is generated by NF-kB derived from ROS delivered by vascular sources controlled by Cyp1b1 or uncoupled turnover of Cyp1a1. The effective activation of AhR in the Cyp1b1-AhR BM signaling by BP may physiologically be replaced by natural tryptophan derivatives from the microbiome that are effective AhR activators [17,20,21]. Adherence signaling distinguishes DKO(+) and DKO(-) within the same BM-MSC cells. Polr2A delivers both DKO(+) and DKO(-) genes that are distinguished by adherence signaling to DKO(+).

A novel non-canonical PAH activation of AhR is shown for these Cyp1b1-TCDD clusters. First, the AhRd allele is as effective with BP as the AhRb allele. BP reaches the AhR in BM with particular efficacy in *Cyp1a1*^-/-^ mice. Secondly, DMBA—which almost matches BP in the canonical stimulations of Ahrr, Spint1, and Cyp1a1—is consistently only about 30 percent effective with all Cyp1b1-AhR stimulations [68,69]. Each switch is compatible with replacement of Arnt by an alternative heterodimer partner.

### 2.3. Special Functional Features of Cyp1b1-TCDD Signaling

#### 2.3.1. Cell Adhesion Distinguishes DKO(+) and DKO(-) Clusters: BP-Cyp1a1^-/-^ and Cyp1b1^-/-^ Stimulations Parallel One Another

For the DKO(+) cluster, each stimulation correlated inversely with the suppressed expression in adherent WT cells (Figure 3A). The cells that express these genes became a higher proportion of the assemblages due to losses of abundant lymphocytes, erythroblasts, and osteoblasts (Figure 2E; Appendix A). According to our model (Figure 1A), these stimulations by Cyp1b1 deficiency in DKO(+) genes include a constant underlying cell enrichment factor of 2- to 3-fold that arises from the loss of other cell types (Figure 1D). This enrichment factor combines with an intrinsic stimulation that connects to adhesion signaling. Large intrinsic stimulations that showed no adherence bias were demonstrated by 12 genes (DKO(+)^NAdh^; Appendix A). DKO(-) genes also demonstrated uniformly high stimulation, but without adherence dependence.

The *Cyp1b1*^-/-^ stimulations are closely matched by the BP-Cyp1a1^-/-^ responses (Figure 3B; Appendix A). Removal of Cyp1a1 metabolism enhances local BP levels [52]; in many respects, BP is modeling effects of endogenous Cyp1a1 substrates perhaps provided by the microbiome, which reaches the BM from the gut epithelium.

The contribution from Cyp1a1 metabolism to the stimulation following Cyp1b1 deletion indicates a novel crosstalk between these Cyps. The extra deletion of Cyp1a1 in the DKO mice substantially decreases the Cyp1b1^-/-^ effect in the DKO(+) sub-cluster and completely removes the stimulation in the DKO(-) sub-cluster. Several questions arise concerning the location and substrates for this basal Cyp1a1 intervention. Notably, basal Cyp1a1 expression is scarcely detectable in BM. The gut epithelium provides a plausible source that communicates readily with the BM sinusoids [17,18,20]. The joint shared activity implies a mixing of active products from Cyp1b1 and Cyp1a1 in the BM.

When Cyp1a1 is deleted, BP becomes far more active in the stimulation of genes in the DKO(+) and DKO(-) sub-clusters. There is a correlation with *Cyp1b1*^-/-^ stimulations for DKO(+) genes (Figure 3B, Appendix A). Importantly, the stimulations of the same genes in BP-treated DKO mice again correlate, but with a three-times lower response. Moreover, the added BP now scarcely contributes to the stimulation by DKO. The removal of both Cyps may even elevate the circulating BP concentration.

The DKO(+) group included numerous abundantly expressed genes (Figure 3C). Several genes exhibiting stress linkages (Tp53, Chd4, Shc2) were also represented in this group. Others were multi-potential lineage markers (Nestin, Col2a1, Myod1, Sox15). Most multi-potential development markers in the DKO cluster showed a lower expression, which overlapped the low expression of DKO(-) genes.

#### 2.3.2. RNA Polymerase 2 Subunit A Is a Control Factor for Cyp1b1-TCDD Signaling

The most responsive DKO(+) gene was RNA polymerase 2 subunit A (Polr2a), the largest of 12 subunits. In addition, there were correlations with Nestin (Figure 2F) and Olfr1410 (Figure 3G). Polr2a contained both the catalytic and key regulatory sites [41,42]. Substantial correlation with other DKO(+) genes suggested that the availability of this Pol2 subunit controls the DKO(+) cluster (Figure 3D,F, Appendix A). The relative expression levels of the other 11 subunits, across adhesion and *Cyp1b1*^-/-^ groups, were similar. However, subunits E, G, and L were always the highest, including in BMS2 cells, while subunit H was the lowest. In BM, the *Cyp1b1*^-/-^ stimulations brought the A subunit up to the level of the B subunit, which matched the levels of subunits C, D, F, J, and K. The treatment effects on the Polr2 subunits were compared in Appendix A. Polr2a was also correlated with the linker histone, H1fx (Figure 3G), with an anomalously high expression in cluster DKO(+) (Appendix A). In mouse embryos, H1fx is co-expressed during organogenesis with Nestin [70] (Appendix A). H1fx functions to maintain chromatin in a pluripotent state [63].

#### 2.3.3. OLFR Expression and α-Defensin Activation

Ectopic OLFRs are distributed across both DKO(+) and DKO(-) clusters [23,24,71] (Figure 3D). These OLFRs were stimulated by both *Cyp1b1*^-/-^ and BP-Cyp1a1 (Figure 3E; Appendix A). OLFRs respond to small volatile organics that are released by the microbiome (Appendix A) [23,24,72]. Polr2a correlated closely with the most responsive, Olfr1410 (r^2^ = 0.89) (Figure 3G). Cyp4a10, Rgs3, and Rdh12 were each potential OLFR response modulators (Figure 3D). The most abundant DKO(+) gene was α-defensin (Defcr3) [63]. Highly expressed α-Defensin delivery provided a potential function for the OLFR network by delivering polar peptides with potent anti-adhesion and antibiotic activities [73].

#### 2.3.4. OLFR, α-Defensin, and Polr2a Features Extend to BMS2 Cells

To explore whether DKO(+) genes are expressed in cells in the BM-MSC lineage, we examined the expression in BMS2 cells, which are generated from BM-MSC after 14 days [48,74]. Ten OLFR genes and α-defensin each replicated their expression in the adherent BM cells (Figure 3F,H). A high proportion of BM DKO (+) genes retained appreciable expression in BMS2 cells (Figure 3H, Appendix A). BMS2 cells, at high density, also showed exceptionally low Polr2a expression relative to the remaining 11 subunits, although neither AhR activation nor metabolite stress enhanced this low expression [75] (Figure 3F). This overlap with the BM-MSC line strongly suggested that the 65 DKO(+) genes—including the 35 OLFR genes—in the BM-adherent assembly derive from multi-potential cells of the BM-MSC lineage, which became enriched by in vivo Cyp1b1 deletion or TCDD activation of AhR.

### 2.4. Characterization of How Cyp1b1 Can Affect Assembly Cells In Vitro

#### 2.4.1. A Design to Test How Cyp1b1 May Support Lymphoid Progenitors

We have identified the in vivo generation of multiple partnerships between AhR and Cyp1 forms (Figure 2 and Figure 3). Cyp1b1 controls an association of oxygen/ROS with cell adhesion, a key feature of these partnerships. Cyp1b1 can function directly at sites of expression or through secondary paracrine effects. We have established methodology to study the same BM cells ex vivo via maintenance of lymphocyte progenitors [76] or expansion of monocyte lineages to macrophages [48,77]. Cyp1b1 modulation of intercellular ROS signaling is tested in the first, while direct effects within the monocyte lineage are examined in the second protocol. These considerations extend to the Cyp1b1 origin of assemblage gene changes, including lymphocyte dissociation, BM-MSC enrichment, and intrinsic signaling.

Lymphoid progenitors are abundant in both adherent (Adh) and non-adherent (Non-Adh) BM cells (Appendix A). They are highly sensitive to DNA double strand breaks (DDSB) generated by both ROS and PAH dihydrodiol epoxide (PAHDE) metabolites formed from DMBA [78,79,80,81]. To test the ex vivo impact of *Cyp1b1*^-/-^ on B cell progenitors, the BM cells were cultured on an OP9 MSC monolayer [46]. As a mechanism reference, we also used mice lacking XPC, a component of the DDSB repair complex [46] (Figure 4A). We compared the protective effects of Cyp1b1 and XPC during a 24 h culture with OP9 direct contact (OP9), with a 24 h pre-enrichment (OP9 + EM), or with an enriched medium without OP9 cells, with five different conditions (1–5) (EM only, cond-1, 2 and 3). Basal DDSB were enhanced by the extra PAHDE-delivery metabolism in the OP9 monolayer (50 µM PAH) (cond-4 and 5) or by AhR activation (cond-5), which parallels TCDD suppression (direct AhR).

**Figure 4 ijms-24-16884-f004:**
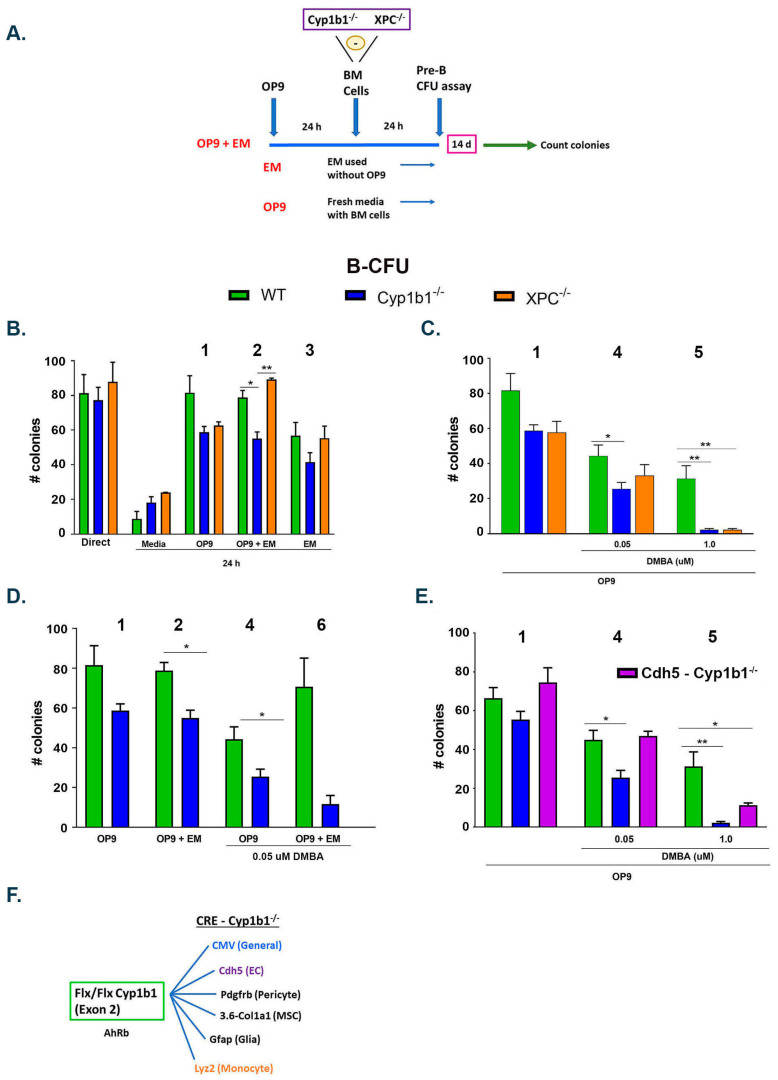
In vivo *Cyp1b1*^-/-^ changes in vitro lymphocyte progenitors through DNA double strand breaks (DDSB). Endothelial-cell-specific Cyp1b1 targeting. (**A**) in vitro support of isolated BMCs using the MSC monolayer (OP9 cells). Lymphocyte progenitors in Pre-B populations are assessed for colony forming activity with a stimulant cocktail (CFU assay), either directly after isolation or after 24 h under various conditions. In CFU assays, 5 × 10^4^ cells were plated and then promoted to form clonal colonies over 14 days. For WT cells, 80 ± 10 colonies form to provide a count of initially active progenitors. These lymphoid progenitors are supported by cytokines released from the OP9 monolayer (including IL7 and Csf1). Medium enrichment (ME, 24 h) releases additional factors (OP9 + ME). ME alone sustains lymphoid progenitors. The effectiveness of in vivo Cyp1b1 is compared to DDSB sensor XPC, with respect to maintenance of lymphoid progenitor proliferation after 24 h of an in vitro culture (B-CFU). Comparisons to activities, and six different experiments were compared numbered 1–6, in parts B, C, D, and E. Numbers refer to comparisons of the same data in different combinations. (**B**) Standard OP9 co-culture deletions of Cyp1b1 and XPC caused similar basal losses attributable to oxygen-induced ROS. Extra EM factors selectively improved protection when XPC was deleted. (**C**) (1 vs. 4) 50 nM DMBA converts to reactive metabolites (PAHDE). CFU losses were similarly enhanced for deletions of either Cyp1b1 or XPC. An increase to 1 µM DMBA (1 vs. 5) produces much greater losses, again equally enhanced for both deletions. The large boost in suppression matches the DMBA concentration range for which DMBA activates AhR [56]. The overlap for Cyp1b1 and XPC, a specific DNA strand repair component, implicates Cyp1b1 in such protection. (**D**) (4 vs. 6). ME delivers factors that protect WT but destabilize Cyp1b1^-/-^ progenitors. (**E**) EC-specific *Cdh*5-Cyp1b1 deletion fails to replicate the suppression produced by 50 nM DMBA (condition 4) but effectively reproduces the response to 1 µM DMBA (condition 5). This selectivity of *Cdh*5-Cyp1b1 deletion confirms that the mechanisms are different. The effect of direct AhR activation by TCDD is matched by 1 µM DMBA and BP [56]. (**F**) Generation of *Cyp1b1^fl/fl^* mice allows selective deletion of exon 2 when bred with mice that express lineage selective Cre endonucleases [44]. CMV-Cre provides systemic deletion. *Cdh*5-Cre deletes Cyp1b1 from endothelial cells (EC), and Lyz2-Cre deletes Cyp1b1 from myeloid lineages. * *p* < 0.05 and ** *p* < 0.01.

#### 2.4.2. Cyp1b1 Lowers Intracellular ROS, and XPC Facilitates DNA Repair in Targeted Cells

OP9 MSC releases cytokines that fully sustain the B cell progenitor expansion assay (B-CFU). The major disruptions derive from ROS that are generated by 20 percent oxygen in the culture. Additional ROS DDSB damage occurred with removal of either Cyp1b1 or XPC (Figure 4B; cond-1). The low expression of Cyp1b1 in isolated BM cells implies effects in the sinusoids prior to their release. The enrichment medium from OP9 cultures (OP9 + EM) had no effect on the loss from *Cyp1b1*^-/-^, but provided an alternative DDSB protection to compensate for XPC deletion (Figure 4B; cond-2). Removal of OP9 cells (EM only) lessened the support, but continued to compensate for XPC deletion (Figure 4B; cond-3).

A low PAH concentration (50 nM) delivers PAHDE from the support OP9 monolayer to the lymphocytes [82] (see Appendix A), supplementing basal ROS disruption. Comparative analyses with Cyp1b1 and XPC deficiency demonstrated the retention of similar 40 percent losses. Thus, Cyp1b1 and XPC appear to be involved in a similar protection (Figure 4C; cond-4). A 20-fold increase in PAH (1 µM) produced only a small further suppression (Figure 4C cond-5 vs. -4), indicative of a plateau level of ROS/PAHDE/DDSB. However, CFU proliferation was almost completely removed, with the deletion of either XPC or Cyp1b1. At this higher PAH concentration, the suppressions in WT cells matched those of TCDD, indicative of parallel mechanisms functioning through AhR activation (Figure 4C; cond-5 vs. Figure 4E). The elevated disruption by 50 µM DMBA was enhanced by the enriched medium (OP9 + EM) (Figure 4D; cond- 6 vs. 4). The XPC^-/-^ cells showed no such enhanced disruption (not shown). We concluded that Cyp1b1 and XPC similarly diminish DDSB, although in different ways. The differences become resolved by medium factors when ROS disruption is enhanced by PAHDE. Cyp1b1 in BM cells lowers medium ROS outside lymphocytes, but not PAHDE delivered from OP9 cells. XPC enhances DDSB repair within cells, including these progenitors.

#### 2.4.3. Cyp1b1 Functions in a Cell-Selective Manner to Lower ROS

Cyp1b1 lowers ROS in multiple cell types within the BM niche, including pericytes, EC, MSC, and monocytes. The *Cyp1b1*^flx/flx^ line [44] is activated by exon 2 deletion when bred with mice expressing the CMV-Cre transgene. This systemic *Cyp1b1*^-/-^ mouse fully reproduces the obesity suppression of the classic exon 3 disruption [44,66,67,68] (Figure 4E). Mouse Cre lines that generate lineage-specific deletions test the cell source of functional Cyp1b1 activities and Cyp1b1-AhR partnerships (Figure 4F). Here, we compared systemic CMV-Cre deletion effects on lymphocyte progenitors with EC-specific (*Cdh*5-Cre; *Cyp1b1*^flx/flx^) deletion. EC and pericyte Cyp1b1 expression suppressed vascular ROS [68,83]. This process synergized with DMBA metabolites (PAHDE) in the targeting of DDSB (Figure 4C,E). EC-specific Cyp1b1 deletion reproduced 1 µM DMBA effects, which we attributed to AhR, but not the 50 nm DMBA/PAHDE suppression (Figure 4E). Deletion of Cyp1b1 at other BM locations (pericytes, MSC) could also deliver the PAHDE effect on B-CFU.

### 2.5. Mechanism of Participation of Cyp1b1 in ROS Activation

#### Cyp1b1 Deletion Selectively Enhances Oxygen-Dependent ROS Signaling in Purified EC That also Activates AhR

N-acetylcysteine (NAC) is a selective inhibitor of ROS signaling [15]. In our OP9 co-culture model, NAC prevented DMBA-mediated B-CFU losses that were enhanced in the *Cyp1b1*^-/-^ cells, but not in *Xpc*^-/-^ BM cells (Figure 5A). These CFU activities may have been resolved by access of NAC to different pools of ROS. The protection is distinguished by location. Cyp1b1 probably functions on ROS external to the vascular cells, which have access to NAC. XPC functions on nuclear DNA, which has less access to NAC. In isolated EC and pericytes, *Cyp1b1*^-/-^ additionally increased Cyp1a1 expression (Figure 5B). Similar oxygen-mediated stimulation of Cyp1a1 in lung epithelia is mediated by AhR [30].

The participation of EC Cyp1b1 in the protection of lymphoid CFU complements previous in vivo BM studies, in which Cyp1b1 in BM-MSC rapidly targets the same cell types as DMBA. In addition, the intervention by ROS-selective NAC emphasizes the functional role of ROS in these processes. This EC induction of Cyp1a1 in concert with *Cyp1b1*^-/-^ implicates ROS with AhR activation. This process potentially models the *Cyp1b1*^-/-^ and Cyp1a1 crosstalk that resolves DKO intervention in the signaling for DKO(+) and DKO(-) clusters.

In these *Cyp1b1*^-/-^ EC, NAC reversed the oxygen-dependent inhibitory effects on EC capillary organization [69,84] (Figure 5C,D). Isothiocyanate inhibitors of NF-κB activation were similarly effective. *Cyp1b1*^-/-^ activation of ROS stimulated IκB kinase, which is targeted by these inhibitors. This stimulation is initiated through DDSB and ATM (ataxia telangiectasia mutated) signaling, which also links to the NF-κB complex [37,38,70,71]. The *Cyp1b1*^-/-^ effect on ROS generation was additionally marked by lipid peroxidation products, identified by increased acrolein staining (Figure 5F). The active NF-κB heterodimer usually comprises Rel A or Rel B and a p50 partner. In EC and pericytes, Rel A functions as the active transcription factor, becoming phosphorylated at S^276^ by Cdk9 and at S^311^ by z-PKC [72] (Figure 5E). The NF-κB activation can partner with AhR to deliver an alternative, non-canonical activation, which is a candidate for this Cyp1b1-TCDD process [8,38,70] (Figure 5G).

**Figure 5 ijms-24-16884-f005:**
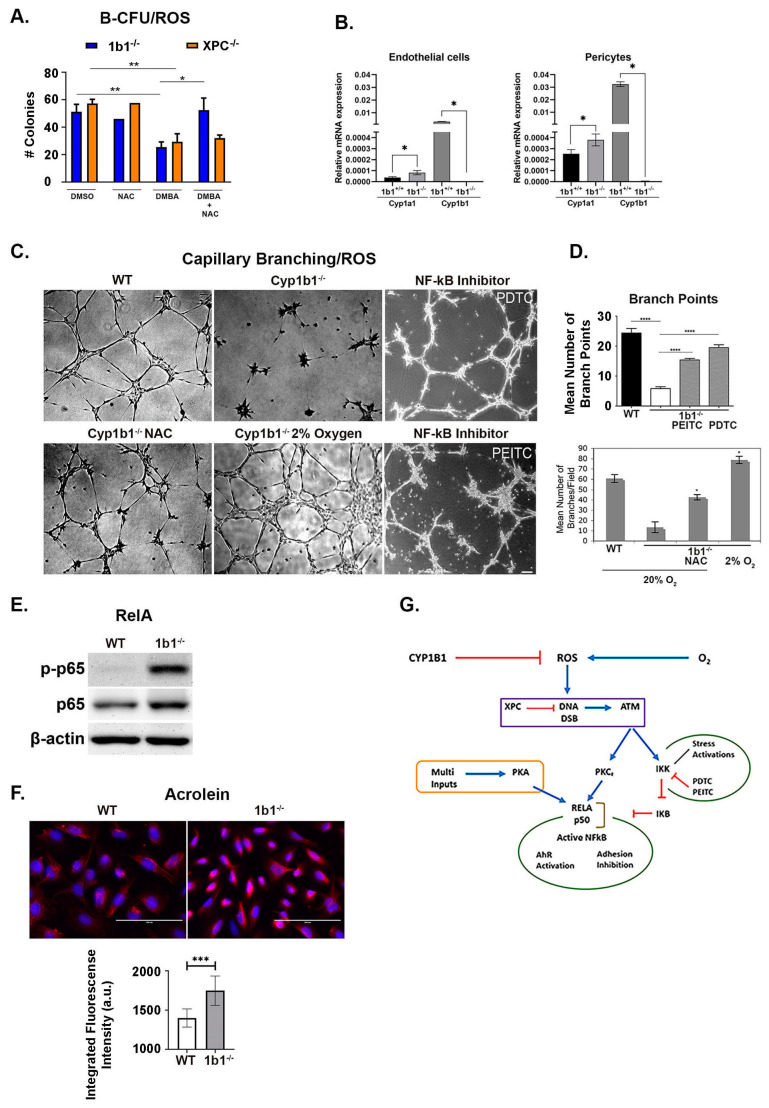
Effects of in vivo *Cyp1b1* deficiency on in vitro cultured endothelial cells (EC). Identification of changes mediated by ROS. (**A**) NAC, a selective ROS inhibitor, reverses *Cyp1b1*^-/-^-directed lipid peroxidation via the PAHDE/ROS combination (blue). *XPC^-/-^* suppression (orange) is not reversed, indicating that residual DNA repair is unaffected. This suppression is independent of EC Cyp1b1 expression (Figure 4E). (**B**) Cyp1b1 deficiency induces Cyp1a1 expression in both EC and pericytes. (**C**) Cyp1b1 deficiency disrupts EC capillary morphogenesis, which is also dependent on oxygen concentration and IκK activation of NF-κB (Thiocyanate inhibition). (**D**) Branch points quantify capillary morphogenesis. (**E**) Cyp1b1 deficiency activates RelA phosphorylation, detected by the p-S311 antibody (PKCz target). (**F**) Cyp1b1 deficiency increases lipid peroxidation (contributor to ROS), quantified by acrolein-linked fluorescence. (**G**) Oxygen-dependent connections between Cyp1b1, ROS, IκK, RelA, and alternative AhR activation. ROS produces DNA strand breaks that activate ATM, which in turn stimulates PKCz to phosphorylate RelA, a non-canonical partner for AhR. Please note panels (**C**) ([84]) and (**D**) ([69]) were prepared from our previous publications with permission. * *p* < 0.05, ** *p* < 0.01, *** *p* < 0.001, and **** *p* < 0.0001.

### 2.6. Cyp1b1 Interventions in Csf1 Expansion of Myeloid Progenitors to Macrophages

#### 2.6.1. Csf1 Expansion Generates Distinct Types of M2 Polarization of M0 Macrophages

ROS-mediated processes play a critical role in inflammatory processes. We have examined the role of Cyp1b1 in macrophages generated from myeloid progenitors that have been detected in myeloid lineages, and have been detected in these mouse BM cells (Figure 1B). Csf1 promotes progression of myeloid progenitors, CMP and GMP, to dendritic cells (DC) and naive M0 macrophages (Figure 1 and Figure 6A). The respective gene responses to Csf1—and subsequently to IL4 and Ifnɣ—resolve 21 monocyte/macrophage PCR markers into seven groups with distinct modulation by Cyp1b1 (Figure 6B; Appendix A).

**Figure 6 ijms-24-16884-f006:**
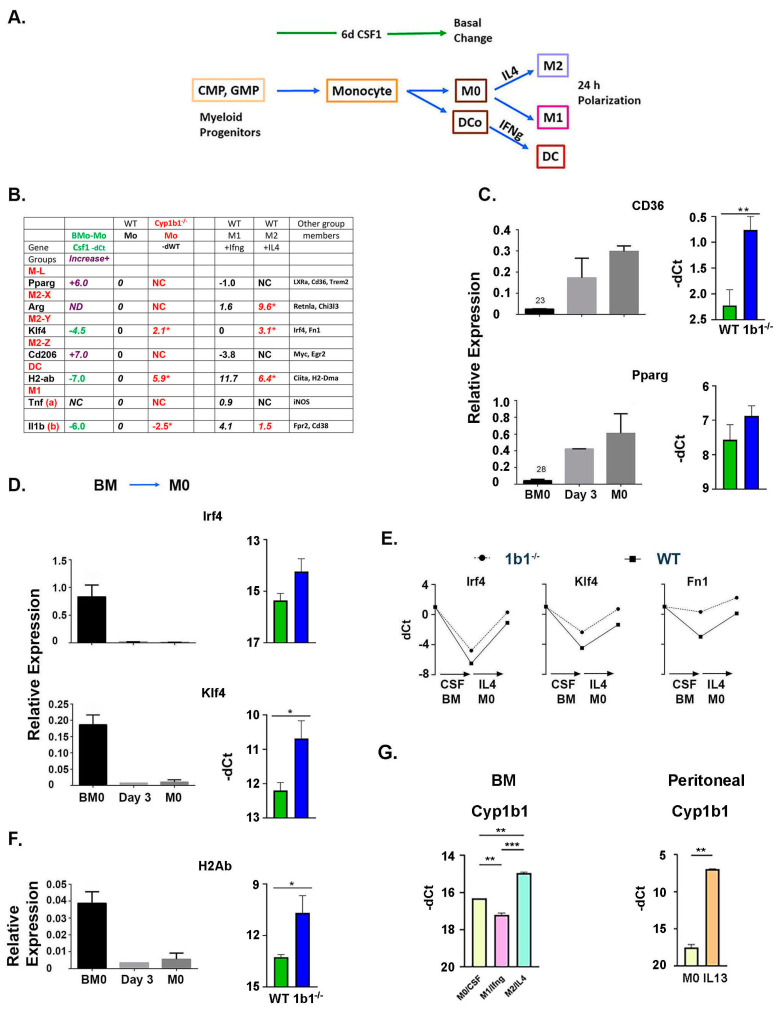
Effects of in vivo *Cyp1b1*^-/-^ on the in vitro expansion of myeloid progenitors to macrophage and dendritic cells (DC). (**A**) Design. Expansion of myeloid progenitors to M0 macrophages and dendritic cells by CSF1. Basal M0 macrophages are assessed through specific polarization stimulations by Ifnɣ (M1) and IL4 (M2), each assessed by qPCR of select marker genes. All expression levels are shown relative to β-actin, either as standardized relative expression or as -dCt, which quantifies increased expression in ln2 expression units. (**B**) Gene markers resolve distinct polarization clusters (**C**,**D**) with selective effects of *Cyp1b1*^-/-^ (**E**). Basal DC are also expanded by Ifnɣ (**F**). **M-L**, Pparɣ and lipogenic genes increase with Csf1 but scarcely after additional IL4. Three M2 polarization clusters. Changes produced by Csf1 from BM expression. **M2-X**, low responses to Csf1. Major stimulations by IL4; **M2-Y**, large suppressions by Csf1. Additive reversals by *Cyp1b1*^-/-^ and IL4; **M2-Z**, large stimulations by Csf1. No further stimulation by *Cyp1b1*^-/-^ and IL4; **DC**, large suppressions by Csf1. Partial reversal by *Cyp1b1^-/-^.* Large reversal by Ifnɣ. (**C**) Csf1 increases M-L markers. *Cyp1b1*^-/-^ only stimulates CD36 relative to WT. ** *p* < 0.01 (**D**) Csf1 decreases M2-Y markers. *Cyp1b1*^-/-^ stimulations partially reverse. * *p* < 0.05. (**E**) *Cyp1b1*^-/-^ and IL4 increase M2-Y genes additively to reverse Csf1 suppression. * *p* < 0.05. (**F**) Csf1 decreases DC marker H2-ab. *Cyp1b1*^-/-^ M0 change decreases Csf1 suppression. (**G**) Cyp1b1 in M0 is suppressed by Ifnɣ but increased by IL4. Basal expressions in BM macrophage and in peritoneal macrophage are similarly low (−17 cycles), but the increase in peritoneal macrophage by IL13 is much larger (11 vs. 2 cycles). ** *p* < 0.01; ** *p* < 0.001.

The ML cluster marked by PPARɣ, response targets such as CD36, and other lipogenic receptors (LXR forms and Trem2) increased during the Csf1 expansion that produced the M0 macrophage. These genes were insensitive to the subsequent M1 polarization provided by Ifnɣ and the M2 polarization provided by IL4 (Figure 6B,C). Cyp1b1 deletion did not affect PPARɣ levels, but appreciably elevated the target gene CD36. The constitutive activity of PPARɣ appears to be stimulated. The M2 polarization reflects a regenerative lipogenic state that is produced by PPARɣ. The effects on the expression of the Csf1 expansion and subsequent treatments by Ifnɣ and IL4 resolved three distinct groups with different M2 formulations marked by Arg1, Klf4, and CD206 (M2-X, M2-Y and M2-Z), respectively (Figure 6B). They are resolved by their polarization responses to Ifnɣ and IL4. Only the M2-Y group (Klf4, Irf4, and Fn1) is changed by Cyp1b1 deletion (Figure 6D,E). Each shows an increased M0 expression following Csf1 expansion that is sustained additively during the IL4 stimulation. The dendritic monocytes (DC) marked by H2Ab showed a similar elevation following Csf1 suppression (Figure 6F).

The expression of Cyp1b1 was low following the Csf1 expansion, suggesting that effects of deletion occur at an earlier stage in the expansion of the myeloid progenitor cells (Figure 6G). The M0 expressing Cyp1b1 showed the same M2-Y polarization as Klf4, Irf4, and Fn1. IL4 increased Cyp1b1 while Ifnɣ caused a decrease. Notably, a previous report [48] showed that an equivalent IL4 stimulation of Cyp1b1 was prevented by the deletion of Irf4. We also found that macrophages isolated from the peritoneum of comparable mice showed high Cyp1b1 expression after an M2 stimulation by IL13. Peritoneal macrophages originate from distinct embryonic sources to populate resident tissue macrophages.

#### 2.6.2. Increased M2 Polarization Is Demonstrated by Decreased Participation of M1-Marker Genes

M1 genes also exhibited heterogeneous responses to Csf1 and IFNɣ. After the Csf1-induced expansion, the basal macrophages showed that Tnf and iNOS did not respond to *Cyp1b1*^-/-^. Alternatively, IL1β and Fpr2 showed basal decreases (Figure 6B and Figure 7A). An additional 24 h challenge with IFNɣ superimposed additively on basal differences, with variable response times from 3 to 12 h (Figure 7B). Each of these transient responses was decreased in *Cyp1b1*^-/-^ BM cells (Figure 7C). Part of the transience of these M1 responses derives from the rapid turnover of the mRNA. The origin of the near complete reversal within 6–12 h remains unknown.

**Figure 7 ijms-24-16884-f007:**
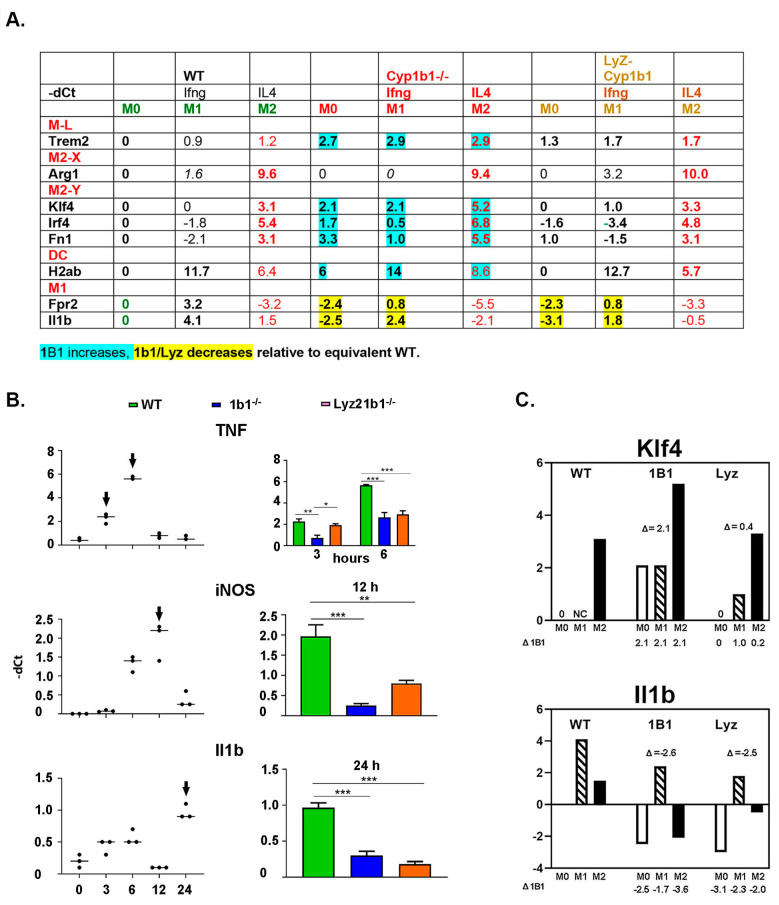
In vivo *Cyp1b1*^-/-^ changes that derive from monocyte-selective Cyp1b1 expression. (**A**) Effects of systemic (CMV) and monocyte (Lyz2) Cre *Cyp1b1*^-/-^ deletions vs. WT. For each gene, WT M0 is set to zero. Changes in -dCt show increases in expression relative to M0 as positive. Twenty-four hour responses to Ifnɣ (M1/DC) and IL4 (M2) are additive with basal M0 shifts. Basal Cyp1b1^-/-^ stimulations (blue) for ML (Trem2), each M2-Y gene, and DC (H2ab are additive with the corresponding Il4 and Ifnɣ stimulations). The equivalent treatments for Lyz2 Cyp1b1 cells failed to produce equivalent responses. Basal and M1/Ifnɣ stimulations were similarly suppressed in Cyp1b1-/- and Lyz2 Cyp1b1 cells (yellow). (**B**) Time courses for Ifnɣ stimulations of M1 markers (Tnf, iNOS, and IL1β) each show early responses within 12 h that are independent of 24 h outcomes (Table A in **A**). Right: Mean peak CMV/Lyz responses to Ifnɣ (n = 3). (**C**) *Cyp1b1*^-/-^ changes basal (M0) but not additional polarization responses. The basal *Cyp1b1*^-/-^ shift is shown in each of the three separate cultures. *Cyp1b1*^-/-^ changes relative to equivalent WT for M0, M1, and M2. Klf4; D1B1 (2.1) is not replicated by Lyz2 *Cyp1b1*^-/-^ (0.4). IL1β loss for *Cyp1b1*^-/-^ (−2.6) is replicated by Lyz2 *Cyp1b1*^-/-^ (−2.5). * *p* < 0.05, ** *p* < 0.01, and *** *p* < 0.001.

### 2.7. Cell Selectivity of Cyp1b1 Responses

Three types of Csf1-stimulated monocytes were each enhanced by systemic *Cyp1b1* deletion (M-L, M-Y, and DC) (Figure 7A-blue), while M1 markers were suppressed (yellow). The M0, M1, and M2 changes that are sensitive to Cyp1b1 deletion were further probed by restricting the deletion to the monocyte lineage (Lyz2-Cre; *Cyp1b1^fl/fl^*) (Figure 4F). The basal increases with systemic Cyp1b1 deletion noted that in the M2-Y trio, the DC MHC2 markers and the lipogenic Trem2 were not retained with the Lyz2-Cre deletion. Thus, Cyp1b1 is functioning for these changes from cells that are outside the monocyte lineage (Figure 7B). However, basal *Cyp1b1*^-/-^ effects on basal M1 responses and on the acute Ifnɣ stimulations were reproduced by the monocyte-restricted Lyz2 Cre-*Cyp1b1^fl/fl^*. Figure 7C compares the expression changes produced by the 24 h control, Ifnɣ, and IL4 treatments in WT, systemic CMV-Cre; *Cyp1b1^fl/fl^*, and monocyte Lyz2-Cre; *Cyp1b1^f^*^l/fl^. The pattern for three separate cultures with M2-favored KLF4 expression showed a Lyz2 pattern that was close to the WT. By contrast, the equivalent three cultures for M1-favored IL-1β show matching responses for systemic and monocyte-selective changes.

## 3. Discussion

### 3.1. Rapid Adherence of Isolated BM Cells Is Sensitized, In Vivo, by Cyp1b1 Deletion and AhR Activation in BM Sinusoids

Isolated BM cells include many cell types from different lineages that progress during subsequent culture [1,2,12]. About 5 percent of the cells adhere to plastic within 60 min, forming multi-cellular assemblages. Changes occur over the next 24 h, without further adherence. The multiple adherent cell types respond extensively and similarly to in vivo Cyp1b1 deletion or AhR activation by TCDD [9,56]. Over 200 genes shared similar responses to these dual challenges (Cyp1b1-AhR cluster). These responses included both changes in intrinsic cell signaling and appreciable redistribution of assemblage cell types.

Here, we provide a model for these heterogeneous assemblages, which initially after isolation include a high proportion of lymphocytes and erythroblasts. We provide evidence that the 200 genes of the Cyp1b1-AhR assemblage were expressed in MSC and became enriched after a 14-day culture as the BMS2 cell line. In vivo, *Cyp1b1* deletion and 24 h AhR activation changed their adhesion, much as previously described for the C3H10T1/2 MSC line [56]. Consequently, lymphocytes and erythroblasts dissociate, differentiation to osteoblasts declines, and MSC become enriched. Macrophages, which form a substantial proportion of the assemblages, change little in vitro. These assemblages contain appreciable numbers of progenitor cells that progress with stimulation by the medium’s factors during a 24 h cell culture, such that MSC expansion occurs after that period [1,2,3,4,5].

Cyp1b1 was scarcely detectable in the adherent cells, despite an extensive in situ histological presence in BM sections [74]. This difference strongly suggests that the adherent cells are sensitized by changes in the sinusoids, in vivo, prior to their release. We provide evidence that Cyp1b1 in macrophages and EC have appreciable effects in vivo, probably through effects on mesenchymal lymphoid and myeloid lineages. EC also have a low presence in the isolated cells, suggesting sinusoidal participation. EC vary according to the tissue of origin [15], but the oxygen-dependent increases in ROS and NF-κB signaling, shown here in purified retinal EC, are typical [68]. Removing Cyp1b1 switches the cells from adherence to proliferation [15]. Such decreases are introduced physiologically, notably by binding the Mir-27b to 3′UTR sites [14].

The Cyp1b1-AhR genes in these adherent BM cells also became maximally elevated when BP-mediated AhR activation was prolonged by the deletion of Cyp1a1. This change removes most BP metabolism. Here, BP can be viewed as a surrogate for physiological tryptophan products from the microbiome. Many are Cyp1a1 substrates that also activate AhR [7,19,20,77]. Constitutive Cyp1a1 is largely absent from BM, but is elevated in the gut epithelium, which connects to BM sinusoids through arterial blood flow (Figure 8) [19,20,77]. Depletion of gut bacteria removes metabolites that otherwise stimulate AhR elevation of Cyp1a1. The ligand and allelic selectivity indicate non-canonical BP/TCDD stimulation of AhR in WT adherent cells [56,77]. Evidence for RelA as a candidate AhR partner for ROS activation is presented [7,67]. Other potential partners include KLF6 and the estrogen receptor [8,37].

The different cell types in the adherent assemblages exhibited a diversity of cell-selective signaling. The pattern of matrix AhR and Cyp1 responses that characterize each of six processes is shown in Figure 2A (also see Appendix A). Direct canonical AhR responses are few (eleven), demonstrating very low expression and possibly limited AhR activity in erythroblasts. Indirect canonical participation, mediated by AhR-induced Cyp1a1 (BP-Cyp1a1 genes), was notable for transient (<12 h) increases of about twenty genes, including several key cytokines typically associated with macrophages and neutrophils. Two other specialized cell-selective mechanisms link to deletions of Cyp1b1 and Cyp1a1, respectively. The first mechanism was notable for association of Cyp1b1 with DDSB, and immunoglobulin generation in B lymphocytes [71]. In addition, deletion of Cyp1a1 targets eosinophils, possibly involving enhanced endogenous AhR activation in the gut epithelium, elevating HSP70 and suppressing particulate ribonucleases (EAR) [7,17].

### 3.2. Cell Signaling Associated with Cyp1b1-AhR Genes

The Cyp1b1-AhR genes are divided into two clusters, distinguished by the extensive loss of stimulation caused by the removal of Cyp1a1 from *Cyp1b1*^-/-^ mice (DKO state). The major cluster (75 genes) retains *Cyp1b1*^-/-^ stimulation in the DKO state [30–40%; DKO (+)]. Many of these genes are highly expressed, with expression patterns that were repeated in the BMS2 cell line, which was developed from BM-MSC cells [9]. This cluster, therefore, marks an MSC type cell. The diminished expression in the WT-adherent assemblage (Negative for Adh/Non-Adh) suggested an exclusion by adhesion signaling. This adhesion effect is reversed after in vivo Cyp1b1 deletion or AhR activation by TCDD. Notably, the stimulations correlate with the original negative effect of adhesion. The smaller DKO(-) cluster (25 genes) lacks basal adhesion suppression and showed high stimulations from Cyp1b1 deletion. However, this expression entirely depends on Cyp1a1, as evidenced from the complete return to basal expressions in the DKO mice.

A clue to the function of the BM-MSC was provided by the abundance of responsive stress markers (Trp53, Chd4, and Shc2). The most highly expressed gene was α-Defensin/Def3a [70]. This protein targets microbial cell membranes and is enriched in Paneth cells that are associated with the gut epithelium [63,79]. A positive interplay between *Cyp1b1*^-/-^ and Cyp1a1 could derive not only from differences in their respective reduction cycles [27,29], but also from the distinctive metabolism of estradiol (E2) [43,78] and polyunsaturated fatty acids [26,79]. Cyp1a1 becomes elevated in *Cyp1b1*^-/-^ EC, as seen in lung epithelia [30]. This stimulation increases the possibility of a contribution to the stimulations by *Cyp1b1*^-/-^ seen in DKO(+) and DKO(-) sub-clusters.

### 3.3. Novel Gene Responses

The two features of the DKO(+) cluster were particularly remarkable. First, gene expression correlated with specific stimulation of the A subunit of RNA polymerase 2 (Polr2a) [41,42], possibly partnered by the inter-nucleosome histone-like protein, H1Fx [63]. The remaining 11 Polr2 subunits, which probably serve epigenetic support functions, remained in fixed ratios. Secondly, each cluster was dominated by re-purposed OLFRs [23,24,25] that comprised approximately 20% of the respective genes. The OLFR genes were expressed with Def3a in BMS2 cells, which also expressed very low Polr2a relative to H1Fx and the other subunits. This OLFR’s presence in both DKO(+) and DKO(-) clusters, as well as in BMS2 cells, suggested expression in lineage-related BM-MSC cells. These GPCRs sense volatile organic molecules generated by microbiome bacteria [13,23,62]. This ectopic expression has been documented in pancreatic islet cells, with appreciable overlap with the present OLFR set (Appendix A) [25]. In the islet cells, octanoic acid, a typical microbiome product, stimulated Olfr15 to enhance glucose sensitivity of insulin release.

H1Fx is an inter-nucleosome binding protein that is found in regions of high Pol2 binding [62]. The very high expression of H1Fx in adherent assemblages suggested a dual presence DKO(+) BM-MSC and was unstimulated in a second abundant adherent cell type, such as erythroblasts. H1fx functions to maintain stem cell character. H1Fx and Cyp1b1 localize to similar regions in mouse embryos [13,62].

### 3.4. Impact of Cyp1b1 Deletion In Vivo on Ex Vivo Progenitor Functions

Deletion of Cyp1b1 produced DDSB via increased oxygen-induced ROS [10,80]. The effective protection provided by Cyp1b1 to lymphocyte progenitors matched that of XPC, a protein that delivers DDSB repair. This dual protection is evident with additional stresses from external PAHDE and AhR activation. This involvement in DSBB regulation in BM explains the exceptional stimulation of immunoglobulins that occurs in the adherent lymphocytes after Cyp1b1 deletion.

The Cyp1b1-dependent suppression of lymphoid progenitor proliferation was also reversed by the ROS inhibitor, NAC, without effect on XPC-mediated processes. Cyp1b1 and NAC control intercellular ROS and cellular redox processes [15]. The alternative DNA repair available in *Xpc^-^*^/-^ cells was apparently sensitive to medium factors, but not these redox processes. The selective removal of Cyp1b1 from EC (*Cdh*5-Cre) specifically countered the extra suppression delivered by high PAH levels and AhR activation. Lower chemical stress from ROS and low PAH levels could be accommodated by other less active Cyp1b1 sources.

The in vitro Csf1 expansion of monocyte lineages also increased PPARɣ and lipogenic genes that were controlled by this receptor. These lipogenic macrophages are similar to those in adipose tissue [73]. Cyp1b1 exhibited a modest M2 polarization, but is expressed at much lower levels in BM macrophages [48] than in peritoneal macrophages from these mice. Cyp1b1 deletion primarily stimulated basal levels of PPARɣ-targeted M2-response genes (Cd36, Irf4, and Klf4), but not other M2 marker genes (Cd206 and Arg1) or their IL4 stimulation. Basal M1 markers and their acute stimulations by INFɣ were, however, suppressed. Specific deletion of Cyp1b1 from the monocyte lineage (*Lyz2*-Cre) exclusively decreased these M1/INFɣ stimulations, thus distinguishing their origin from the PPARɣ responses. Here, vascular ROS activation of PPARɣ was a likely source of Cyp1b1 intervention.

### 3.5. Overview of Cyp1b1 and OLFR Regulation

In Figure 8, we treat the BM and gut epithelium as a single unit connected by arterial blood flow [18]. Cyp1b1 from the different cell types controls multiple active forms of ROS [15,79]. Cyp1b1 also exhibits dual effects on progenitors in the myeloid lineage that are stimulated by sinusoid Csf1 [15,79]. The activity of PPARɣ, which is elevated by Csf1, is enhanced by the removal of Cyp1b1 from outside the lineage. Other M2 polarization processes (CD206, Arg1) are unaffected. EC Cyp1b1 is a candidate source, as evidenced by effects on lymphoid progenitors. Monocyte/macrophage Cyp1b1 (Lyz2-Cre deletion) disrupts Ifnɣ stimulation of M1 polarization.

The AhR may function canonically in the gut epithelium and non-canonically in BM. Csf1 released from EC maintains clonal myeloid progenitor expansions that are parallel to the in vitro model [3]. Crossover regulation of BM-MSC receptors by cytokines that are produced by neutrophils and macrophages have previously been presented [9]. The multiplicity of cell-specific Cyp1b1 contributions to these BM cells needs to be evaluated for selectivity on the Cyp1b1-AhR genes. OLFR-mediated GPCR signaling plays a critical role in connecting diet and microbiome to inflammatory and immune functions.

### 3.6. Final Perspective

Cyp1b1, which was originally discovered from the mouse embryonic multi-potential C3H10T1/2 cell line, is also highly expressed in a BM counterpart multi-potential BMS2 cell line. Cyp1b1 and BM-MSC are low in isolated WT assemblages, despite a strong presence in bone sections. While the proportion of adherent cells changes little in the initial 24h, the low MSC proportions may increase several fold without detection such that the cultures become self-sustaining and enriched in BM-MSC within 72 h [12]. Cyp1b1 has a dominant presence matching the BMS2 cell line within 2 weeks.

Post-pubertal control of BM development is age-dependent [85]. Growth hormone (GH), a prime BM regulator, functions through Igf1. Adult circulating GH differs between males (pulsatile) and females (slow fluctuations). GH signaling to the liver is strongly linked to Cyp1b1 [66]. These BM assemblages lose their Cyp1b1 responsiveness as they age. Other BM signaling changes dramatically between 8 and 52 weeks [85]. The goal here has been to use precisely defined conditions to identify novel BM-MSC and clusters of co-regulated DKO(+) and DKO(-) genes that are Cyp1b1- and AhR- dependent under the defined conditions used here. The next phase focuses on how other conditions affect these characteristics.

In vivo, Cyp1b1 in BM-MSC generates PAH metabolites that suppress sinusoidal lymphoid and myeloid progenitors within 4h [9]. Cyp1b1 also controls oxygen-dependent adhesion and proliferation of vascular cells [15], perinatal liver development [13], and inflammatory monocyte functions [48]. Substrates include estradiol and ω3-fatty acids [43,79] as local modulators, probably directed by oxygen levels [9,15].

Deletion of Cyp1b1 in BM engages rapidly eluted adherent BM-MSC in a non-canonical partnership with AhR (Cyp1b1-AhR cluster) that far exceeds local canonical effects. Selective Cre-deletions of *Cyp1b1^fl/fl^* from respective endothelial and myeloid cell sources resolves effects on associated BM lymphoid and myeloid progenitors. However, the substrate mediators remain to be identified. Multiple genes in the Cyp1b1-AhR cluster overlap with genes expressed in BMS2 cells. Included are many OLFRs that potentially respond to volatile products from microbiome digestion [23,24,25]. Constitutive AhR activation is generated by microbiome tryptophan metabolism. Each may transfer from gut arterial BM blood flow [17,18,20,22] (Figure 8). These Cyp1b1-AhR clusters are correlated with stimulations of the catalytic subunit of RNA polymerase 2. Eleven other subunits remained constantly high [41,42]. Participation of constitutive Cyp1a1 resolves sub-clusters based on the replacement of *Cyp1b1*^-/-^ by DKO [DKO(+) and DKO(-)].

Cyp1b1 matches DDSB repair gene XPC in the protection of DNA in lymphoid progenitors, thus complementing work on ROS suppression. Cyp1b1 may similarly influence myeloid progenitors [11]. Monocyte Cyp1b1 sustained M1 macrophages despite minimal expression. However, the high expression in peritoneal macrophages [26], which derive from embryonic progenitors, potentially points to an early lineage intervention. Please also see Appendix A that focuses on pharmacology/functional details of clusters and Appendix A.

## 4. Materials and Methods

### 4.1. Animals

C57BL/6J (wild type: WT), Rosa26 Flp recombinase expressing (B6.129S4-Gt(ROSA)26Sortm1(FLP1)Dym/RainJ; Stock#: 009086), CMV-Cre (B6.C-Tg(CMV-Cre)1Cgn/J; stock#: 006054), Lyz2-Cre (B6.129P2-*Lyz2^tm1(cre)Ifo^*/J; stock#: 004781), Cad5-Cre (B6.Cg-Tg(*Cdh*5-cre)7Mlia/J; stock#: 006137), and *XPC* knockout (KO: XPC^-/-^; B6;129-*XPC^tm1Ecf^*/J; stock#: 010563) mice were purchased from the Jackson Laboratories (Bar Harbor, ME, USA).

### 4.2. Generation of Conditional Knockout Mice

Targeted Cyp1b1tm1a(KOMP)Wtsi Premium ES cells were obtained from the KOMP Repository (UC-Davis). Chimeric mice were generated in the University of Wisconsin Genome Editing and Animal Models Core Facility. Briefly, Clone G08 was expanded and subsequently microinjected into blastocysts from C57BL/6J mice. Injected embryos were transferred into pseudo-pregnant recipients, resulting in seven chimeric mice. Cyp1b1Fxneo progenies were mated with C57BL/6J mice for 5 generations in the University of Wisconsin Biomedical Research Model Services Core Facility. F1 heterozygote progeny were mated with the Rosa26 Flp recombinase-expressing mouse to remove the neomycin selection marker. The resulting heterozygote littermates (Cyp1b1^flx/+^) were mated to generate the *Cyp1b1^flx/flx^* mice. The floxed mice were further backcrossed for 10 generations. The male floxed mice were subsequently bred with female CMV-cre mice for the generation of the new global Cyp1b1-CMV-cre deletion mice. Genotyping for the FLP recombined allele, intact LoxP sites, and Cyp1b1 Cre-recombined alleles was confirmed by Transnetyx Genotyping [44].

VE-cadherin/Cyp1b1-ko were generated from crossing a *Cyp1b1^flx/flx^* mice with a VE-cadherin (*Cdh*5) Cre, resulting in an endothelial specific *Cyp1b1*^-/-^. The Lyz2-Cre excision was completed by breeding the *Cyp1b1^flx/flx^* mice with the Lyz2-cre mice that had been back-crossed into the C57BL/6J background for 10 generations. The established animal strains were bred and housed in our animal care facilities at the AAALAC certified University of Wisconsin Madison School of Veterinary Medicine and Biotron Animal Care Units, and used in accordance with the NIH Guide for the Care and Use of Laboratory Animals.

### 4.3. In Vivo Chemical Treatment

PAHs, TCDD, or a vehicle control (olive oil) were administered to the mice via a single intraperitoneal injection (DMBA, 50 mg/kg; BP, 50 mg/kg; TCDD, 30 µg/kg). PAH and TCDD were purchased from Accustandard (New Haven, CT, USA). Mice were euthanized and bone marrow was isolated 6-, 12-, or 24-h post-injection, as indicated.

### 4.4. Bone Marrow Isolation

Mice were euthanized under CO_2_ anesthetization. Femurs were extracted and BM cells isolated via flushing or Type 1 collagenase [10] (Worthington, Lakewood, NJ, USA) digestion of crushed bone material in medium (RMPI with 2% FBS) for 20 min at 37 °C. Fresh BM-MSCs were filtered through a 40 micron filter, washed with a medium, and MSC progenitor complexes were allowed to adhere to plastic over 60 min at 37 °C. Cells from one mouse were plated to 1 well of a Corning 6-well dish. The resultant adherent fraction was washed 3 times with culture medium prior to cell collection via trypsinization.

### 4.5. Microarray mRNA Profiling

Microarray analyses were completed in accordance with previously published studies [9,10,11]. Microarray expression data were analyzed from untreated WT mice, as well as mice treated with TCDD (12 h) and BP (6-, 12- and 24-h) for comparison to *Cyp1b1*^-/-^, *Cyp1a1^-/-^* and the combined double knockout (DKO) mice (Figure 2A).

### 4.6. PCR Analyses of Macrophages

RNA was isolated from BM-MSC by the standard Trizol and Qiagen RNeasy Mini kit procedures. cDNA libraries were generated using random hexamers and GoTaq polymerase (Promega, Madison, WI), according to the manufacturer’s protocol. Expression levels of 20 to 25 genes were determined by qPCR using primer pairs shown in Appendix A. For each gene, dCt shifts relative to β-actin were determined, as well as relative expression levels from a serial dilution standard curve.

### 4.7. Cell Culture

OP9 cells were purchased from ATCC (Manassas, VA, USA). Cells were cultured under standard conditions (37 °C at 5% CO_2_ in saturated atmospheric humidity), in a 10% FBS-supplemented medium. Feeder stromal cultures were grown to approximately 100% confluence and treated with 10 µg/mL mitomycin C (Sigma, St. Louis, MO, USA) for 24 h to inhibit further cellular division. Mitomycin C (2 mg/5 mL) socks were maintained in PBS and kept frozen in liquid nitrogen. Treated feeder layers were maintained in culture for up to 6 weeks. Co-cultures were comprised of freshly isolated primary bone marrow cells on a treated feeder layer. OP9-enriched medium (EM) was defined as the medium recovered from feeder layer cultures alone after 24 h. All PAH exposures were tested after 24 h incubation, unless otherwise noted. Incubations were completed with separate EM, and with the combination of OP9 monolayer and EM (OP9 + EM).

### 4.8. PAH Treatment of Cultured Cells

Freshly isolated primary BM-MSC were plated at a density of 1.05 x 10^6^ cells/cm^2^ in either fresh culture medium (αMEM with 20%FBS) or OP9-EM, prepared as described above. Co-culture experiments were completed using feeder cells, while monocultures were completed with primary BM cells on tissue-culture-treated plastics. DMBA or DMSO control treatments were added to the cell suspensions at plating. All treatments were 24 h in culture prior to re-collection of the BM cells for analysis. Figure 1A depicts treatment strategies for each of the groups tested by colony forming unit (CFU) analysis.

### 4.9. Colony Forming Unit Assays

CFU assays were completed as per the manufacturer’s instructions. In brief, BM cells were isolated directly from femur extraction/bone marrow isolation, or after 24 h chemical treatment (Figure 1). The BM cells were resuspended in a fresh culture medium—5 × 10^5^ cells/mL for pre-B progenitors (CFU-B). Cells (1/10 volume) were mixed with the appropriate MethoCult medium (Stem Cell Technologies, Vancouver, BC, Canada) and incubated at standard culture conditions (37 °C at 5% CO_2_ in saturated atmospheric humidity) for 7 days. Colonies were counted by visual inspection via light microscopy.

### 4.10. CSF1 Expansion of Macrophages and Polarization

BM cells, isolated from crushed femurs by collagenase digestion, were plated at a density of 5 × 10^5^ cells/mL in RPMI 1640 with 10% FBS, supplemented with L-glutamine, penicillin/streptomycin, a non-essential amino acid mix, and β-mercaptoethanol [48]. The culture was stimulated with M-CSF at 20 ng/mL. On day 3, the medium was renewed for a further 3 days with MM/Csf. On day 6, the medium was re-stimulated with MM/Csf for a further 24 h to provide Mo macrophage. Flow cytometry (F4/80) indicated the macrophage content had increased from 25% to 85% percent. M1 macrophages were obtained using MM plus 20 ng/mL INFɣ for this 24 h period. M2 macrophages were obtained using MM plus 20 ng/mL IL4.

Gene expression, determined after M0, M1, and M2 treatments, was completed with RNA isolated from duplicate BMC cultures. Cells were isolated from single groups of mice with different genotypes [WT, *Cyp1b1*^-/-^, (Cmv-Cre *Cyp1b1^fl/fl^*), and Lyz2-Cre *Cyp1b1^fl/fl^*]. Time courses during the 24 h incubations were completed with intermediate times (3 h, 6 h, 12 h, 24 h). A second experiment compared separate WT and *Cyp1b1*^-/-^ mice in triplicate.

### 4.11. Mouse Peritoneal Macrophages

The generation of mouse peritoneal macrophages followed procedures previously reported for studies of Cyp1 forms in these cells. Male C57BL/6J mice, 5–8 weeks of age, were used. After sacrifice, peritoneal macrophages were washed from the peritoneal cavity with 5 mL of 0.9% NaCl. Collected cells were centrifuged (1500 rpm × 10 min) and suspended in a DMEM medium supplemented with glutamine, penicillin, streptomycin (all from Invitrogen, Waltham, MA, USA), and 5 percent heat inactivated fetal calf serum. Cells were allowed to adhere for 2 h at 37° and 5% CO_2_. A further wash with PBS removed non-adherent cells. The adherent cells were stimulated with IL-13 (20 ng/mL) for 24 h. mRNA was isolated and assessed for multiple genes by PCR, as described.

### 4.12. Retinal Endothelial Cells and Pericytes Studies

Retinal endothelial cells (EC) and pericytes (PC) from wild-type (*Cyp1b1*^+/+^) and *Cyp1b1*^−/−^ mice were isolated and maintained as previously described [68,84]. Cells were cultured in their appropriate growth medium containing 10% fetal bovine serum and 5.7 mM D-glucose at 33 °C with 5% CO_2_. In order to compare Cyp1a1 and Cyp1b1 mRNA expression levels in cells, total RNA was extracted from *Cyp1b1*^+/+^ and *Cyp1b1*^−/−^ retinal EC and PC using a combination of TRizol reagent (15596026; Invitrogen, Waltham, MA, USA) and RNeasy mini kit (74104; Qiagen, Valencia, CA). The cDNA synthesis was performed using 1 μg of total RNA using an RNA to cDNA EcoDry Premix kit (639549; Clontech, Mountain View, CA, USA). Real-time quantitative PCR (qPCR) analysis was performed in triplicate using TB-Green Advantage qPCR Premix (639,676; Clontech) with specific primers. Target genes were normalized using the simultaneous amplification of ribosomal protein L13A (Rpl13a), a housekeeping gene as previously described in [81,86]. qPCR primer sequences are as follows: *Cyp1a1* forward: 5′-TCCATACATGGAAGGCATGA-3′, *Cyp1a1* reverse: 5′-TCTTTTGGGAGGAAGTGGAA-3′, *Cyp1b1* forward: 5′- TCCAGCTTTTTGCCTGTCAC-3′, *Cyp1b1* reverse: 5′-TGGCTGGGTCATGATTCACA-3′, Rpl13a forward: 5′- TCTCAAGGTTG TTCGGCTGAA-3′, and Rpl13a reverse: 5′-GCCAGACGCCCCAGGTA-3′.

The ability of *Cyp1b1*^+/+^ and *Cyp1b1*^−/−^ retinal EC to form a capillary-like network on Matrigel (10 mg/mL, 354,234; BD Bioscience, Bedford, MA) with different conditions was evaluated as previously demonstrated [69,84]. Tissue culture plates (35 mm) were coated with Matrigel and incubated at 37 °C for at least 30 min to solidify the Matrigel. Cells were incubated with specific inhibitors for 8 h, and 1 × 10^5^ cells were plated in a serum-free growth medium with specific inhibitors on the top of the solidified Matrigel. For hypoxia conditions, Matrigel-coated plates and the growth medium were pre-equilibrated in the hypoxia chamber (2% O_2_, 5% CO_2_, and 93% N_2_) for 24 h. After being plated on the Matrigel, cells were incubated for 18 h and photographed using a phase microscope. For quantitative analysis, the mean numbers of branch points were determined by counting the number of branch points in 5 representative high-power fields (×100). Reagents used in this study were as follows: phenethyl isothiocyanate (PEITC, 1 μM, 253731; Sigma, St. Louis, MO, USA), pyrrolidine dithiocarbamate (PDTC, 10 nM, P8765, Sigma), and N-Acetyl-L-cysteine (NAC, 1 mM, A9165; Sigma).

To assess NF-κB activity in cells, western blot analysis of phosphorylated NF-κB subunit RelA (p-p65) and p65 was performed as previously described [81]. Cell lysates (25 µg) were separated by SDS-PAGE using 4–20% Tris-Glycine gel (XP04202; Invitrogen, Waltham, MA, USA) and transferred to nitrocellulose membranes (10,600,001; Cytiva, Marlborough, MA, USA). Membranes were blocked with 5% skim milk prepared in Tris-Buffered Saline containing 0.05% Tween-20 (TBST, BP337-500; Thermo Fisher, Hanover Park, IL, USA) and incubated with primary antibodies (1:1000) overnight at 4 °C. Primary antibodies were as follows: anti-phosphorylated-NF-κB p65 (3033; Cell Signaling, Danvers, MA, USA) and anti-NF-κB p65 (8242; Cell Signaling). The membranes were washed with TBST and incubated with horseradish peroxidase-conjugated secondary antibodies (1:3000, Jackson ImmunoResearch, West Grove, PA, USA) at room temperature for 1 h to detect protein bands using ECL Western Blotting Detection Reagents (RPN2209; Cytiva, Marlborough, MA, USA). The same blot was re-probed with an anti-β-actin antibody (MA5-15739; Thermo Fisher, Hanover Park, IL, USA) as the loading control.

To evaluate lipid peroxidation levels in cells, indirect immunofluorescence analysis was performed to compare levels of acrolein, a product of lipid peroxidation reactions. *Cyp1b1*^+/+^ and *Cyp1b1*^−/−^ retinal EC were plated on glass coverslips coated with 5 µg/mL of fibronectin (354,008; Corning, Steuben, NY). The cells were fixed with 4% paraformaldehyde for 15 min on ice and permeabilized with 0.1% Triton X-100 in phosphate-buffered saline (PBS, D1408; Sigma) for 10 min at room temperature. The cells were blocked with 1% of bovine serum albumin (BP9703; Thermo Fisher, Hanover Park, IL, USA) in TBS for 1 h, and stained with an anti-acrolein antibody (1:200, MA5-27553; Invitrogen, Waltham, MA, USA) overnight at 4 °C. The cells were incubated with Cy3-coujugated anti-mouse IgG (1:1000, 715-165-151; Jackson ImmunoResearch, West Grove, PA, USA) for 1 h at room temperature. The cells were washed with TBS, incubated with DAPI (1:2000, D1306; Invitrogen, Waltham, MA, USA) for one minute, and mounted on glass slides using a Fluoromount-G mounting solution (0100-01; SouthernBiotech, Birmingham, AL, USA). The cells were photographed with a Zeiss Fluorescence microscope (Axiophot, Zeiss, Germany) equipped with a digital camera. For quantitative assessment of the data, fluorescence intensities were measured using ImageJ software version 1.52j (NIH, Maryland, MD, USA) and averaged at least 5 images per group.

### 4.13. Statistics

#### 4.13.1. Application of Limma Analyses Large Gene Number/Small Repeat Data Sets

In these array assessments, we used a two-channel display for each array gene site. Each sample contains Cy3-labeled RNA isolated from variously treated adherent cells that compete for the site with a Cy5-labeled reference RNA derived from an equal mix of the three WT non-adherent cell isolations [53]. This internal Cy5-standard normalizes for variations in the arrays and labeling procedures. The processing compares the Cy3/Cy5 normalized ratio at each of 45,000 array sites. Mostly, comparisons have been made for triplicate groups that comprise differences in the mouse genotype or in vivo treatment TCDD or PAHs.

Statistical evaluation of the effects of different treatment groups at each array site are tested through the EDGE^3^ platform, which employs several built-in algorithms [53] that are integrated with Bioconductor, an R-based open-source statistical genomics collaboration. This approach focuses on the LIMMA processing of a full matrix of expression values [54]. Each row represents a gene, and each column corresponds to an RNA expression ratio (Cy3/Cy5) derived from a treatment group. The approach leverages the highly parallel nature of genomic data to estimate different levels of variability between genes and samples. This extension improves reliability when treatment sample numbers are small and effectively clusters gene expression signatures.

The LIMMA approach analyzes experiments in their entirety rather than being limited to pair-wise treatment comparisons. The Empirical Bayes method borrows information delivered by trends within the whole 45,000 microarray set. Thus, the global variance allows for increases in variance as expression is lower. LIMMA also includes a background correction function that improves reads at low levels of expression that dominate these measurements. The LIMMA approach has appreciably expanded in currently available formulations [54].

#### 4.13.2. CFU and PCR Analyses

Statistical significance was determined by ANOVA with a Tukey *post-hoc* test for multiple comparisons; *p* < 0.05, trending was defined as 0.05 < *p* < 0.1 (GraphPad Prism, San Diego, CA, USA).

## Figures and Tables

**Figure 8 ijms-24-16884-f008:**
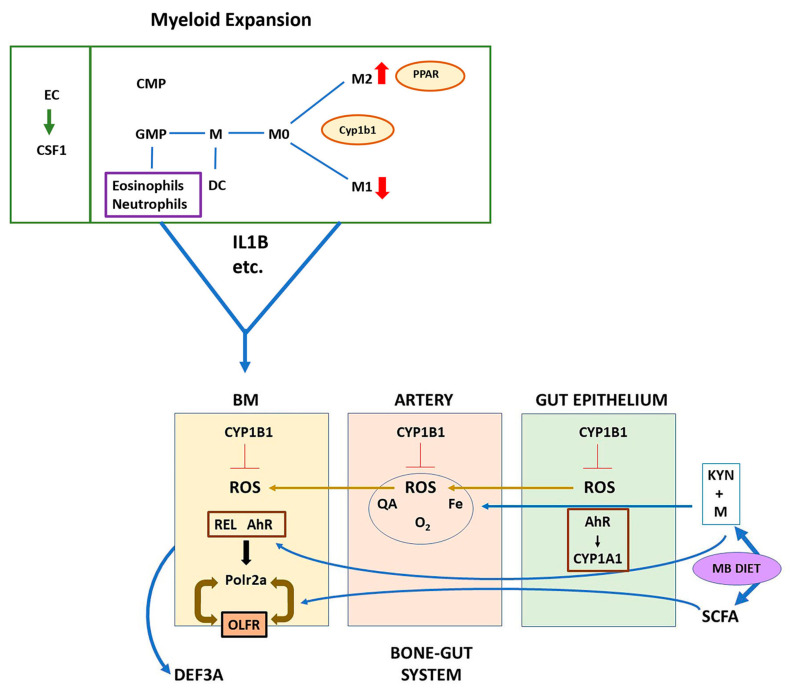
Model for Cyp1b1-AhR partnership between monocytes and MSC in BM sinusoids. (Upper) Local monocyte lineage effects: Csf1 released from sinusoidal vasculature stimulates myeloid expansion to macrophages (M0, M1and M2) that deliver further BM control. Macrophages and monocytes (M), including neutrophils (N) and eosinophils (E), deliver IL1β and other cytokines that prime BM-MSC (Appendix A) in the BM. *Cyp1b1*^-/-^ does not affect total M0 macrophage generation or IL4 stimulations of Arg1 or Cd206 markers. Cyp1b1^-/-^ produces two effects, distinguished by Lyz2-Cre: (i) enhanced activation of PPARɣ (CD36, Klf4, Irf4) (Lyz2^(-)^); (ii) suppression of Ifnɣ stimulation of M1 markers (IL1β, Tnf). Consequently, local cytokine effects on MSC are diminished. (**Lower**) External MSC regulation. Effects of Cyp1b1, Cyp1a1, and AhR in the gut epithelium integrate with regulation in BM sinusoids through connecting arterial blood flow. In BM, Cyp1b1 deletion functions through Cyp1b1-AhR regulation of BM-MSC, which produces DKO(+) and DKO(-) cluster gene responses that are mediated by altered levels of the A subunit of RNA Polymerase 2 (Polr2a) relative to fixed elevated levels of eleven other subunits (Polr2 subunits B to L). OLFR signaling in BM provides an adaptive response to microbiome metabolites from protein and carbohydrate oxidation products, notably short chain fatty acids (SCFA). This OLFR signaling complements responses to tryptophan/kynurenine metabolites that activate AhR in BM cells to form non-canonical complexes with products of ROS/NFκB signaling (for example RelA). This signaling across the gut epithelium is modeled here by BP-*Cyp1a1*^-/-^.

## Data Availability

All the data presented here are included in the manuscript. Further inquiries should be directed to the corresponding authors.

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
