# Peer review of "AhR and CYP1B1 Control Oxygen Effects on Bone Marrow Progenitor Cells: The Enrichment of Multiple Olfactory Receptors as Potential Microbiome Sensors"

_ijms, 2023, doi:10.3390/ijms242316884_

Round 1

Reviewer 1 Report

Comments and Suggestions for Authors

The article “AhR and CYP1B1 control oxygen effects on bone marrow progenitor cells: Enrichment of multiple olfactory receptors as potential microbiome sensors” represent very good piece of work in many respects.

There are the studies very well designed and the experiments described with details. Discussion also took into account all aspects, which should be considered. Statistical analysis of the results was performed perfectly with EDGE platform and LIMMA processing.

The Figures deserve special attention. They are not only of good quality, but also are very well described in Figure legends, what allow readers to know the results before detailed reading of the main text.

There is well written Introduction as well as Discussion.

Summing up, the only my doubt concerns the length of this article. However, rather the division into two articles would be suggested than shortening the current version.

Reviewer 2 Report

Comments and Suggestions for Authors

This manuscript from Larsen et al. is rather difficult to read and understand. One issue is that the data presentation is unclear, many abbreviations and symbols are not defined, and/or information is otherwise missing, contributing to the reader’s difficulties. Changes that would improve readability include: (1) adhering to the convention that descriptions of published results are in the present tense with the results of the current study described in past tense, (2) greater precision in language usage. For instance, the authors talk about Cyp1b1-/- causing effects but this is a shorthand way of expressing the experimental manipulation. Instead Cyp1b1 promotes the opposite effect to that observed, such that deleting the gene encoding Cyp1b1 results in the effect observed. Also, most of the results seem to be more correlative than causative but the authors use strong language indicating mediation rather than association, (3) definition of all abbreviations and symbols and provision of additional information in the figure legends. For example, what is “nc” and “TD” and “BP”? Also, what does DKO(-) and DKO(+) signify? Also, all chemical abbreviations (e.g., TCCD, DMBA, etc.) should be defined on first use, (4) more information on data presentation; for instance, in Figure 2B and D what is shown in the tables? Fold control? What do negative values represent? The authors need to provide more information or preferably find a graphical way to represent these data (such as the Volcano plot shown in panel C).

Major points:

(1) The authors emphasize that in vivo treatment of mice with the various AhR ligands or knockout of Cyp1b1 alters adherence of bone marrow cells but never really explain the significance. What does it mean that the number of adherent cells is altered and why is it important?

(2) The authors need to better explain their matrices, which conditions are being compared to which other conditions, and what the values represent (and how they are calculated).

(3) In Figure 4 what do the numbers above the various columns mean? Do they represent different experiments? If the number is the same in the different panels, are the results from the same experiments? Why are only some experiments analyzed—is it because not all conditions were run in all experiments? Or are the authors “cherry-picking” the results that they analyze? This figure is very unclear.

(4) In Figure 6 what are the numbers above the BM0 bars? In Panel C why is there a -dCt in the middle of the righthand panels not really associated with any axis? In the legend for this figure it is not clear what lines 618 to 623 are referring to.

(5) The authors need to analyze cumulative results from multiple experiments and indicate the n for each figure.

(6) The authors should provide a concise concluding paragraph.

Minor points:

(1) In the Figure 1 legend parts “(i)”,”(ii)” and “(iii)” are mentioned but there is no corresponding labeling in the figure itself.

(2) In Figure 2A what do the “+”, “++”, etc. symbols mean?

(3) There is a mismatch between the legend for Figure 3 and the actual figure.

(4) In line 157 the authors should describe what the changes in steady state levels of progenitor cells are/were.

(5) What does “DKO(+)” and “DKO(-)” mean?

(6) In Figure 7 it is difficult to read the values in black or red font that are highlighted with a dark color.

(7) What does “Cyp1b1 exhibits a modest M2 polarization…” mean? How can an enzyme show macrophage properties? This is an example of imprecise language usage that was discussed in point #2 in the paragraph above.

Comments on the Quality of English Language

The English language is imprecise, which makes the concepts difficult to understand.
